# SARS-CoV-2 RBD trimer protein adjuvanted with Alum-3M-052 protects from SARS-CoV-2 infection and immune pathology in the lung

Nanda Kishore Routhu[1,2,12], Narayanaiah Cheedarla[1,2,12], Venkata Satish Bollimpelli[1,2,12], Sailaja Gangadhara[1,2,12], Venkata Viswanadh Edara [1,3], Lilin Lai[1,3], Anusmita Sahoo[1,2], Ayalnesh Shiferaw[1,2], Tiffany M. Styles[1,2], Katharine Floyd [1,3], Stephanie Fischinger [4], Caroline Atyeo [4], Sally A. Shin [4], Sanjeev Gumber[5], Shannon Kirejczyk[5], Kenneth H. Dinnon III[6], Pei-Yong Shi [7], Vineet D. Menachery [8], Mark Tomai[9], Christopher B. Fox[10], Galit Alter [4], Thomas H. Vanderford[1], Lisa Gralinski [11], Mehul S. Suthar[1,2,3] & Rama Rao Amara [1,2✉]

There is a great need for the development of vaccines that induce potent and long-lasting protective immunity against SARS-CoV-2. Multimeric display of the antigen combined with potent adjuvant can enhance the potency and longevity of the antibody response. The receptor binding domain (RBD) of the spike protein is a primary target of neutralizing antibodies. Here, we developed a trimeric form of the RBD and show that it induces a potent neutralizing antibody response against live virus with diverse effector functions and provides protection against SARS-CoV-2 challenge in mice and rhesus macaques. The trimeric form induces higher neutralizing antibody titer compared to monomer with as low as 1μg antigen dose. In mice, adjuvanting the protein with a TLR7/8 agonist formulation alum-3M-052 induces 100-fold higher neutralizing antibody titer and superior protection from infection compared to alum. SARS-CoV-2 infection causes significant loss of innate cells and pathology in the lung, and vaccination protects from changes in innate cells and lung pathology. These results demonstrate RBD trimer protein as a suitable candidate for vaccine against SARS-CoV-2.

[1] Emory Vaccine Center, Division of Microbiology and Immunology, Yerkes National Primate Research Center, Emory University, Atlanta, GA, USA. [2] Department of Microbiology and Immunology, Emory School of Medicine, Emory University, Atlanta, GA, USA. [3] Department of Pediatrics, Division of Infectious Diseases, Emory University School of Medicine, Atlanta, GA, USA. [4] Ragon Institute of MGH, MIT and Harvard, Cambridge, MA, USA. [5] Division of Pathology, Yerkes National Primate Research Center, Emory University, Atlanta, GA, USA. [6] Department of Microbiology and Immunology, University of North Carolina, Chapel Hill, NC, USA. [7] Department of Biochemistry and Molecular Biology, The University of Texas Medical Branch, Galveston, TX, USA. [8] Department of Microbiology and Immunology, The University of Texas Medical Branch, Galveston, TX, USA. [9] 3M Corporate Research Materials Laboratory, Saint Paul, MN, USA. [10] Infectious Disease Research Institute, Seattle, WA, USA. [11] Department of Epidemiology, University of North Carolina, Chapel Hill, NC, USA. [12]These authors contributed equally: Nanda Kishore Routhu, Narayanaiah Cheedarla, Venkata Satish Bollimpelli, Sailaja Gangadhara. ✉email: ramara@emory.edu

The newly emerged coronavirus SARS-CoV-2, the causative agent of the COVID-19 pandemic has impacted the socio-economic balance worldwide. As of March 30, 2021, SARS-CoV-2 has infected nearly 128 million people resulting in 2.7 million deaths worldwide. Thus, there is an urgent need for the development of vaccines that elicit high titers of long-lasting protective humoral and cell-mediated immune responses and prevent SARS-CoV-2 infection. Recent studies using mRNA, viral vector, protein, and DNA-based delivery platforms have shown that vaccines that induce a strong neutralizing antibody response against the viral spike protein can provide protection in animal models and humans[1–3]. While more than 50 vaccine candidates are currently in a clinical trial, there are only three vaccines (two mRNA-based and one chimp adenovirus-based) that have been approved for human use[4–7]. While great progress has been made in developing vaccines that can induce a strong neutralizing antibody response against SARS-CoV-2, it is not yet clear about the durability of humoral immune response induced by these vaccines, which is critical for ending the pandemic[8]. Towards this end it is important to develop immunogens that induce high titer neutralizing antibody response and combine them with adjuvants that are known to induce long-lived humoral immunity.

The majority of COVID-19 vaccines developed so far employ the spike protein as the antigen to generate protective immune responses against SARS-CoV-2[9]. Spike protein is a major virus surface glycoprotein that engages the interaction with human angiotensin-converting enzyme 2 (hACE2). Spike binds to hACE2 through its receptor-binding domain (RBD) and facilitates virus entry into target cells[9,10]. On the other hand, the S2 subunit facilitates fusion of viral envelope with cellular membrane through the participation of heptad repeat 1 (HR1) and heptad repeat 2 (HR2)[9]. Importantly, most of the neutralizing antibodies generated following SARS-CoV-2 infection and vaccination target the RBD region, and therefore, RBD protein is a promising target to design candidate vaccines[9,11]. Notably, the subunit vaccines developed using prefusion stabilized full-length SARS-CoV-2 spike (S) glycoprotein in combination with saponin-based Matrix-M™ adjuvant showed induction of strong neutralizing antibodies and protection against SARS-CoV-2 in macaques and humans[12–14]. In addition to this, monomeric RBD adjuvanted with aluminum hydroxide also induced neutralizing antibodies against SARS-CoV-2 virus in immunized mice, rabbits, and non-human primates and protected in vivo after SARS-CoV-2 challenge[15].

Accumulating evidence suggests that multimerized antigens are better in engaging interactions with B cell receptors thereby facilitating generation of high-affinity antibodies compared to monomeric antigens[16–18]. Multimerization of either RBD protein or prefusion-stabilized spike (S) glycoprotein using disulfide-linkages, respectively, have been shown to induce higher neutralizing antibody responses than their unmodified versions[2,19]. Several strategies employing similar approaches to multimerize antigens have been shown to enhance humoral immune responses to target pathogens. These include the SpyCatcher-SpyTag system, self-assembling protein nanoparticle immunogens, and several other strategies successful in generating immune responses in preclinical settings[20–22]. A significant advantage of employing multimeric antigen display approach is that they enrich antibody responses to specific epitopes on the target protein and induce stronger neutralizing antibody responses with lower binding antibodies targeting undesirable epitopes. Thus, it is important to consider multimeric display of SARS-CoV-2 spike or RBD protein as vaccine candidates.

Besides immunogen design, adjuvants play a key role in inducing high titer and long-lived antibody response that can provide long-term protection[23]. Currently, there are only a few adjuvants such as alum, MF59, AS03, AS04, AS01, and CpG 1018 that have been approved for human use and some promising adjuvants are currently being tested for safety and immunogenicity in humans[23]. One such example is 3M-052, a synthetic TLR-7/8 agonist, which is a small molecule with an 18-C fatty acyl chain and belongs to the family of imidazoquinolines. 3M-052 is a lipid-modified immune response modifier and was designed to form depots of the adjuvant for a gradual and sustained release for incorporation in oil-in-water emulsions or liposomes. The intercalation of the lipid tail of 3M-052 into the bilayer of the liposome is intended to result in a type of virus-like particle displaying the adjuvant pharmacophore on its surface. 3M-052 maintains local adjuvant activity and does not generate systemic responses evident with non-lipidated structures such as R848[24]. In addition, this modification made 3M-052 more amenable to incorporation in lipid-based formulations such as nanosuspensions, liposomes, or emulsions. By utilizing this property, to create a formulation that is more amenable for human use, Infectious Disease Research Institute (IDRI) created a formulation where 3M-052 was adsorbed onto aluminum hydroxide (alum-3M-052) with the help of a phospholipid excipient. Recent HIV-1 vaccination studies using alum-3M-052 with HIV-1 clade C-derived gp140 immunogen (Env) (1086.C) in rhesus macaques showed that 3M-052 adjuvant generates strong bone marrow (BM)-resident long-lived plasma cells (LLPCs) and increases the magnitude and durability of antibody responses[25]. Consistent with this, we have recently shown that 3M-052 encapsulated in PLGA nanoparticles can induce strong neutralizing antibodies against HIV and provide protection against a simian and human immunodeficiency virus challenge in macaques[26,27]. The positive results in macaques led to the HIV Vaccine Trials Network currently testing the immunogenicity of alum-3M-052 adjuvant in conjunction with HIV envelope trimer protein in humans. These highly encouraging data strongly support testing the utility of alum-3M-052 as an adjuvant for COVID-19 vaccine.

Here, we evaluated the safety, immunogenicity, and protective efficacy of RBD trimer vaccine adjuvanted with alum (a licensed adjuvant) or alum-3M-052 in mice and rhesus macaques. Mouse studies demonstrated that the RBD trimer protein vaccine elicits higher neutralizing antibody response against SARS-CoV-2 virus compared to RBD monomer. Adjuvanting the RBD trimer with alum-3M-052 induced potent neutralizing antibody response and provided complete protection from an infection that was superior compared to alum. Further, studies in rhesus macaques with RBD trimer adjuvanted with either alum or low dose alum-3M-052 induced SARS-CoV-2 neutralization titers with a median of 800 and protected from virus replication in the lower respiratory tract following intranasal and intratracheal (IT) SARS-CoV-2 challenge. In addition, vaccination reduced lung pathology with no or low anamnestic innate, T cell, and antibody responses compared to control macaques. These data support the utility of RBD trimer protein as a potential vaccine candidate against SARS-CoV-2 infection.

## Results

**RBD trimer protein binds to ACE2 with 500-fold higher affinity compared to RBD monomer.** To express a trimeric form of RBD protein we fused amino acid residues 319–541 of SARS-CoV-2 spike protein to the trimerization domain of T4 (Fig. 1A). To promote the secretion of the protein we added the tissue plasminogen activator (tPA) signal sequence at the N-terminus. This fusion protein was expressed under the control of CMV promoter with intron A. We confirmed the expression of the fusion protein using flow cytometry (Fig. 1B) in 293T cells

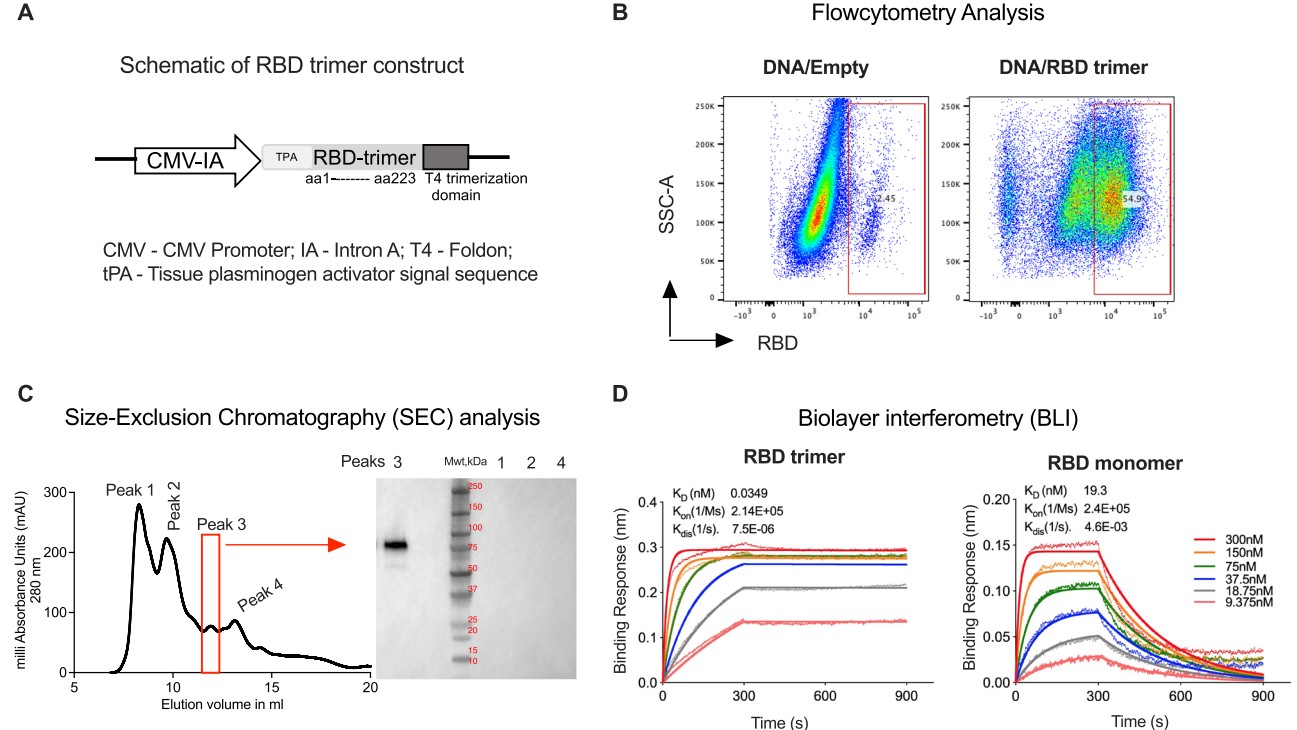

**Fig. 1 RBD trimer protein binds to ACE2 with 500-fold higher affinity compared to RBD monomer. A** Schematic representation of DNA/RBD trimer (pGA8/RBDtri). DNA/RBD trimer encodes for SARS-CoV-2 RBD from amino acid residues 1–223 (represents amino acid residues 319–541 of SARS-CoV-2 spike protein) with the T4 trimerization domain at C-terminus and tPA at N-terminus. **B** Representative flow plots showing the expression of RBD trimer by DNA/RBD trimer. **C** Size-exclusion chromatogram showing RBD trimer peak (left) and western blotting analysis (right) confirms RBD trimer protein expression by plasmid construct. **D** Bio-Layer Interferometry (BLI) sensograms of the binding of DNA/RBD trimer versus monomer to immobilized Fc-human ACE2. The traces represent BLI response curves for SARS-CoV-2 proteins serially diluted from 300 to 9.375 nM, as indicated. Dotted lines show raw response values, while bold solid lines show the fitted trace. The data was globally fit using a 1:1 binding model to estimate binding affinity. The experiments related to **B**–**D** were repeated twice and representative data are shown.

transfected with DNA. We then expressed the protein in 293F cells using the transient transfection with DNA and purified from the supernatants by first binding to Con A agarose followed by size-exclusion chromatography (SEC). We found four peaks on SEC and determined peak 3 to be the RBD trimer based on expected size and Western blotting under non-reducing conditions (Fig. 1C). The typical yield of the protein was around 2.2 mg/liter which can be greatly improved by using a stable cell line constitutively expressing the protein and tag-based purification system. To determine the proper folding of the trimer we measured its binding to human ACE2 using biolayer interferometry (BLI) (Fig. 1D). As a control, we used RBD monomer protein. We observed that RBD trimer binds to ACE2 with very high affinity (KD of 0.035 nM) and this was about 500-fold higher compared to RBD monomer (KD of 19 nM) (Fig. 1D). These results demonstrated that RBD trimer protein fused to T4 trimerization domain formed trimers, folded properly, and binds to hACE2 with very high affinity.

**RBD trimer adjuvanted with Alum-3M-052 induces a robust neutralizing antibody response in mice**. We next determined the immunogenicity and protective potential of RBD trimer protein in BALB/c mice (Fig. 2A). For these experiments, we compared two adjuvants, alum and alum-3M-052. In the initial experiments we vaccinated mice with 10 μg of protein on weeks 0 and 4, and measured antibody responses on week 3 (prime) and week 6 (boost). Following prime, the alum-3M-052 group induced RBD-binding antibody with a titer of $1.7 \times 10^4$ in serum (Fig. 2B). However, these responses were close to the level of detection in the alum group. Following the boost, these titers

increased by 100-fold in both groups with titers reaching $1.7 \times 10^6$ in the alum-3M-052 group and $1.9 \times 10^4$ in the alum group (Fig. 2B). These results demonstrated that RBD trimer adjuvanted with alum-3M-052 induced a robust binding antibody response following two immunizations that is 100-fold higher compared to the response induced by alum. Importantly, the RBD trimer-induced antibody also showed robust neutralizing activity against the live SARS-CoV-2 virus with a geometric mean of 7000 (range 1010–17,628) in the alum-3M-052 group (Fig. 2C). Consistent with a lower RBD-binding titer, the neutralizing antibody titer in the alum group was 100-fold lower with a geometric mean of 66. The RBD binding titer correlated directly with the neutralizing antibody titer (Fig. 2D). The ratio of IgG2a to IgG1 of RBD binding antibody at 2 weeks post-boost was significantly higher in the alum-3M-052 group compared to the alum group demonstrating a Th1-biased response in the former compared to the latter (Fig. 2E).

**RBD trimer adjuvanted with Alum-3M-052 induces higher neutralizing antibody response compared to RBD monomer and protects from SARS-CoV-2 infection in the lungs of mice**. We next compared the immunogenicity of RBD trimer and monomer using the two adjuvants. For these experiments, we used 1 μg of protein, since high doses of protein can mask differences in immunogenicity. Following prime, the low titer of RBD-binding antibody (geometric mean of $10^3$) was observed only in the trimer/alum-3M-052 but not in trimer/alum and monomer with either adjuvant (Fig. 2F). Following boost, the binding antibody titers were high in (trimer and monomer) alum-3M-052-adjuvanted animals but not in alum-adjuvanted animals

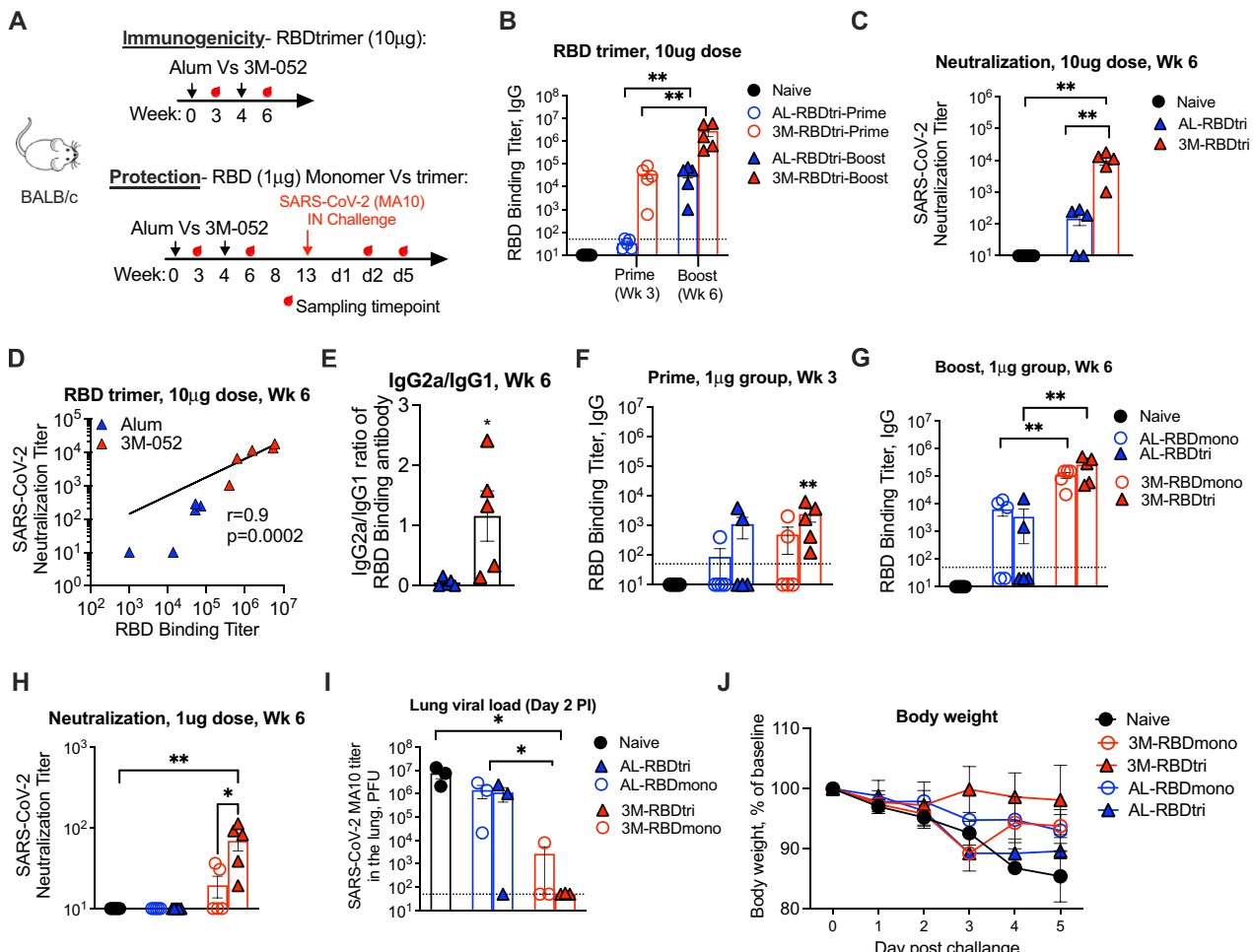

**Fig. 2 RBD trimer adjuvanted with Alum-3M-052 induces a robust neutralizing antibody response and provides protection against SARS-CoV-2 challenge in mice. A** The experimental schedule representing the timeline for the immunizations (study 1) and SARS-CoV-2 (MA10) challenge (study 2). Each immunization group has five animals (n = 5 mice per group). Study 1 (**B**–**E**): 6-week-old female BALB/c mice were immunized with RBD trimer (10 μg/mice) either with Alum (blue) or with 3M-052-Alum (red) via intramuscular route at weeks 0 and 4. **B** Endpoint IgG titers against SARS-CoV-2 RBD measured in serum collected at week 2 post-prime and week 2 post-boost immunizations. Each sample was analyzed in duplicates. **C** Neutralization titer against live mNeonGreen SARS-CoV-2 virus. Each sample was analyzed in duplicates. **D** Correlations between neutralization titer and ELISA binding titers of RBD trimer antigen either with Alum (blue) and 3M-052 (red) adjuvants. The Spearman rank test was used to perform correlation analysis. **E** Bar graph shows the ratio of SARS-CoV-2 RBD-binding IgG2a and IgG1 antibody measurements. Each sample was analyzed in duplicates. Study 2 (**F**–**J**): Six-week-old female BALB/c mice were immunized either with RBD trimer or RBD monomer (1 μg/mice) adjuvanted either with Alum (blue) or with 3M-052-Alum (red) via intramuscular route at weeks 0 and 4. Immunized mice were mock-infected, infected with $10^5$ PFU SARS-CoV-2 MA10. **F**–**G** Endpoint IgG titers against SARS-CoV-2 RBD measured in serum collected at week 2 post-prime (**F**) and week 2 post-boost (**G**) immunizations. Each sample was analyzed in duplicates. (**H**) Neutralization titer against live mNeonGreen SARS-CoV-2 virus. Each sample was analyzed in duplicates. Circles indicate week 2 post-prime and filled triangle indicates week 2 post-boost immunizations. (**I**-**J**) Lung SARS-CoV-2 (MA10) viral titers of vaccinated animals compared to unvaccinated (**I**) and percent starting weight (**J**). In both studies, the black, blue, and red circles indicate naïve controls, alum, and alum-3M-052, respectively. Data are mean ± SEM. A two-sided Mann–Whitney U-test was used to compare between groups, *p < 0.05 and **p < 0.01.

(Fig. 2F). Further, the binding titer in the trimer/alum-3M-052 group (geometric mean of $1.7 \times 10^5$) was 2-fold higher (not significant) compared to monomer/alum-3M-052 group ($8 \times 10^4$). However, the neutralization titer against the live virus was mostly observed in the trimer/alum-3M-052 group and this was significantly higher than all other groups including monomer with alum-3M-052 (Fig. 2H). To determine the protective potential, we challenged mice intranasally with a mouse-adapted SARS-CoV-2 MA10 virus[28] and measured viral titer in the lungs at day 2 (3 out of 5 mice in each group) and weight loss until day 5 (2 out of 5 mice in each group) (Fig. 2I, J). As expected, all control animals showed a very high viral titer of about $10^7$ pfu/lung on day 2 and exhibited weight loss from days 3 to 5. Impressively, all mice in the trimer/alum-3M-052 group were completely protected from

infection and did not show weight loss. 2 out of 3 mice in the monomer/alum-3M-052 group were also protected but showed a transient weight loss on day 3. In contrast, all but one animal in the alum-adjuvanted group was infected, showed high virus replication, and showed some weight loss. These results demonstrated that low dose (1 μg) RBD trimer protein adjuvanted with alum-3M-052 induces neutralizing antibodies against SARS-CoV-2 and provides protection against intranasal SARS-CoV-2 challenge in mice.

**RBD trimer protein induces both neutralizing and nonneutralizing antibodies with effector function and Th1-baised CD4 T cells in macaques.** To further investigate the translational potential of RBD trimer protein as a vaccine for COVID-19, we

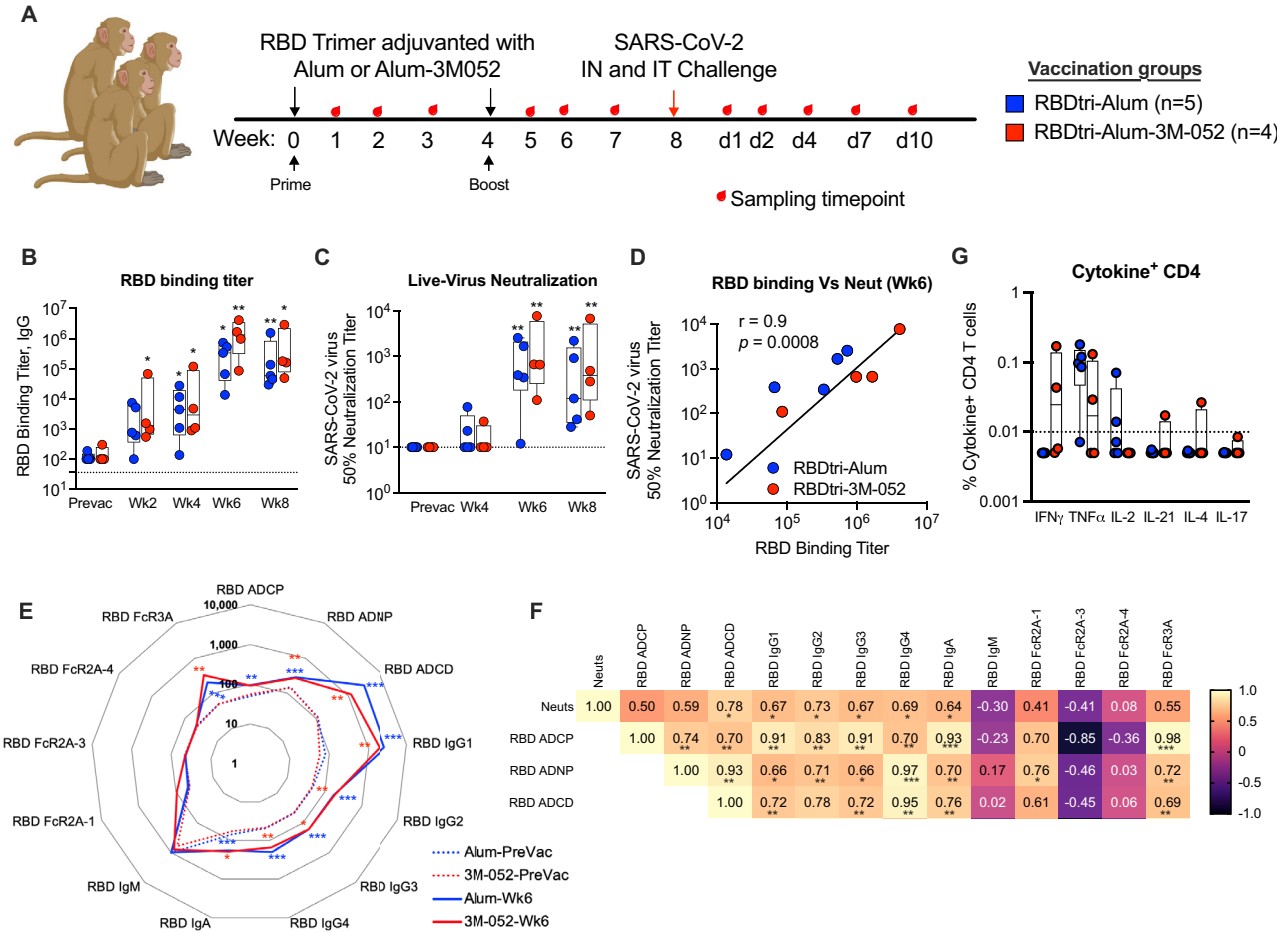

**Fig. 3 RBD trimer induces both neutralizing and non-neutralizing antibodies with effector function in macaques. A** Schematic showing timeline of RBD trimer vaccination, SARS-CoV-2 challenge, and sample collection for the macaque study. Rhesus macaques ($n = 9$) were distributed into two experimental groups to immunize with RBD trimer (30 μg/dose) adjuvanted either with alum (blue) ($n = 5$) or with alum-3M-052 (red) ($n = 4$) via intramuscular route at weeks 0 and 4. Seven unvaccinated animals served as controls. Vaccinated and control animals were challenged intranasally (IN) and intratracheally (IT) with SARS-CoV-2 at 4 weeks following boost. **B** Endpoint IgG titers against SARS-CoV-2 RBD measured in serum collected before vaccination (Prevac) and at weeks 2, 4, 6, and 8 after priming immunization. Each sample was analyzed in duplicates and repeated twice. **C** Neutralization titer against live mNeonGreen SARS-CoV-2 virus in serum. Each sample was analyzed in duplicates and repeated twice. **D** Correlation analysis between SARS-CoV-2 neutralization titer and RBD binding titers. The Spearman rank test was used to perform correlation analysis. Dotted lines reflect the limit of detection. **E** Radar plot showing SARS-CoV-2 RBD-specific systems serology measurements (antibody isotypes and subclass, FcγR binding, and antibody-dependent effector functions) at pre-vaccination and week 6 post-vaccination time points. **F** Correlation matrix analysis between antibody-dependent functions (neutralization, ADCP, ADNP, and ADCD) and SARS-CoV-2 RBD-specific antibody isotypes, and FcγR binding at 2 weeks post the boost. The color refers to $r$ value scale (−1 to 1) shown on the right. The number in each cell indicates the actual $r$ value and the stars represent $p$-values. **G** IFNγ+, TNFα+, IL-2+, IL-21+, IL-4+, and IL-17+ CD4+ T cells specific to total S protein (sum of response to peptide pools S1 and S2) in blood at week 5 post boost. Box and whiskers (min to max) plot showing all the data points in each group. Individual animals were indicated with circles. $p$-values were calculated using a Mann–Whitney test, represent the difference between pre-vaccination and respective post-vaccination value. *$p < 0.05$; **$p < 0.001$; and ***$p < 0.0001$.

immunized two groups of rhesus macaques intramuscularly with 30 μg of protein on weeks 0 and 4 (Fig. 3A). One group ($n = 5$) was adjuvanted with alum (1 mg) and the other ($n = 4$) with alum-3M-052 (1 mg alum, 10 μg of 3M-052). The 10 μg dose of 3M-052 was significantly lower than the dose of 70 μg that has been used in monkeys so far[25]. We used the 10 μg dose since an ongoing human study with HIV envelope is testing 5 μg. Two weeks following prime, low titers of RBD-binding antibody were observed in both groups that were either maintained or marginally increased by 4 weeks (Fig. 3B). However, in contrast to what was observed in mice, the responses were comparable between the two adjuvant groups. This could be due to the use of a lower dose of 3M-052 compared to previous studies that showed superior humoral immune responses by 3M-052

compared to alum[25,29,30]. Following the boost at week 6 (Fig. 3C), both groups showed significantly higher neutralizing antibody titers against live SARS-CoV-2 with geometric mean values of 367 and 778 in the alum and alum-3M-052 groups, respectively compared to controls. Of the 9 animals, 3 animals had a titer >1000 and there was no difference between the two groups, presumably due to the small group size. By week 8 (day of the challenge), the neutralization titer was maintained (<2-fold decline) in 5 animals and declined by 2.5-fold in 3 animals and 9-fold in one animal. At this time, the titer in the alum-3M-052 group was 2.4-fold higher compared to the alum alone group, though not significantly different. As observed with mouse sera, the RBD-binding titer correlated directly with SARS-CoV-2 neutralization titer (Fig. 3D).

We next compared various parameters including IgG subclasses, IgA, IgM, binding to different FcγRs, and functions such as antibody-dependent cellular phagocytosis (ADCP), antibody-dependent neutrophil phagocytosis (ADNP), and antibody-dependent complement deposition (ADCD) (Fig. 3E, Fig. S1). As expected, the IgG response was the most dominant, IgA responses were present at very low levels and the IgM response was negative. Among the IgG subclasses, IgG1 was the most dominant and IgG2, IgG3, and IgG4 were present at low levels. Among the effector functions tested, the ADCD activity was high and the ADCP and ADNP activities were present at very low levels. The vaccine-induced antibody also showed greater binding to FcγR3A and no binding to FcR2A-1, FcR2A-3, and FcR2A-4. The neutralizing activity showed a significant correlation with spike-specific IgG subclass and IgA binding, and ADCD activity (Fig. 3F). Similarly, all three effector functions correlated with IgG-binding activity and with each other. As with the binding and neutralizing activities, there was no difference between the two vaccine groups. Overall, these results show that RBD trimer protein-induced antibodies possess neutralizing activity with diverse effector functions. We evaluated the T cell responses induced by vaccination by stimulating peripheral blood mononuclear cells with overlapping peptide pools specific to RBD protein and intracellular cytokine staining assay (Fig. 3G). After the 2nd immunization we observed low frequencies of CD4 T cells producing IFNγ, TNFα or IL-2 but not IL-4 and IL-17 suggesting a Th1-biased response. However, there was no difference between the two groups. We did not observe any detectable spike-specific CD8 T cell response.

**Alum-3M-052 induces better innate activation compared to alum in macaques following vaccination.** The innate immune system plays an important role in modulating the immune responses mediated by adjuvants. Therefore, we measured the early innate immune responses in the blood of vaccinated animals in terms of changes in cell frequencies (Fig. 4A, B) and their activation based on CD86 expression (Fig. 4C) at Day 1 post primary vaccination, as we showed previously that most changes in DC frequencies and activation occur at this time point following vaccination with alum or TLR7/8 agonist R848[29]. We focused our analyses on three subsets of monocytes (classical, intermediate and non-classical), NK cells, three subsets of DCs (plasmacytoid, BDCA-1+ and CD11c+) and two subsets of myeloid-derived suppressor cells (MDSC) (polymorphonuclear, PMN-MDSC and monocytic, M-MDSC). Vaccination with alum and alum-3M-052 induced differential responses. In particular, we observed that the frequency of classical monocytes decreased and plasmacytoid DC (PDC) increased in alum-3M-052 group compared to alum group, while their pre-vaccination frequencies were comparable. Frequency of M-MDSC increased and NK cells decreased post-vaccination in both the groups, while the decrease in NK cells was profound in alum-3M-052 group compared to alum group.

With respect to activation, alum-3M-052 caused higher activation of intermediate monocytes, PDC, BDCA-1+ DC, MDC, and PMN-MDC (only a trend) whereas alum primarily activated PDC. In addition, the activation of classical monocytes decreased in the alum group. Overall, alum-3M-052 showed activation of monocytes and myeloid DCs post-vaccination while both adjuvants showed activation of PDC. While there was no significant difference in the antibody titer between alum and alum-3M-052 adjuvanted groups in the short term, it is conceivable that the superior innate activation observed in the alum-3M-052 group could lead to induction of long-lived antibody responses in this group.

**RBD trimer protein vaccination provides protection from SARS-CoV2 infection and replication in the lungs of macaques.** To evaluate vaccine efficacy, we challenged vaccinated rhesus macaques with SARS-CoV-2 via intranasal and IT routes at week 8 (4 weeks after the boost). Seven animals that did not receive SARS-CoV-2 vaccine served as controls. Following intranasal and IT challenge we monitored for sub-genomic viral RNA to measure replicating virus in the lung (BAL) and nasopharynx on days 2, 4, 7, and 10 (day of euthanasia) (Fig. 5). On Day 2, 6 out of 7 controls tested positive for virus in BAL (Fig. 5A, B), and 4 out of 7 tested positive in the nasopharynx (Fig. 5D, E). The only animal (RGa17) that scored negative for virus in BAL showed very high virus replication in the nasopharynx ($2 \times 10^7$ copies/ml) indicating that all animals were productively infected. The viral RNA levels were variable and ranged from $2 \times 10^2$ to $2 \times 10^7$ copies/ml. By Day 4, the viral titer decreased in some animals and increased in others and showed a decline by Day 7 in all animals. On Day 10, 2 out of 7 controls still scored positive for virus in BAL and 1 out of 7 controls scored positive for virus in the nasopharynx. In contrast, 3 out of 5 alum vaccinated, and 3 out 4 alum-3M-052 vaccinated animals scored negative for virus in BAL at all times tested. By Day 10, all vaccinated animals tested negative for virus in BAL (Fig. 5B). The area under the curve (AUC) of viral load between Day 0–10 showed significantly lower viral load in vaccinated animals compared to controls (Fig. 5C). A similar protection from infection or viral control was also observed in the nasopharynx of vaccinated animals except that AUC was not significantly different compared to control animals (Fig. 5D–F). These results demonstrated that RBD trimer protein vaccination provides protection from SARS-CoV-2 infection and/or replication in the lungs. We performed correlations between Day 2 viral RNA titer in the BAL or nasopharynx with various antibody functions at the peak of vaccine response (2 weeks post the 2nd protein). We found moderate ($r > 0.5$, $p < 0.1$) inverse correlations (better protection) with neutralizing antibody titer, RBD/spike specific ADCP, ADCD, IgA, and FcR3A-binding activities suggesting that multiple antibody functions contributed to the protection and viral control (Fig. S2).

To assess lung pathology, we analyzed multiple regions of upper, middle, and lower lung lobes at euthanasia. Lung pathologic analyses and scoring (considering severity and number of affected lobes) were performed by two independent pathologists in a blinded fashion. Consistent with early virus control in the lungs vaccinated animals also showed lower lung pathology scores compared to control animals (Figs. 5G, S3, S4, Table S1). The pathology score in the alum-3M-052 animals was significantly lower compared to controls, however, in the alum group the difference was not significant. In addition, animals in the 3M-052-Alum group showed a trend towards lower lung pathology compared to animals in the Alum alone group. Overall, these data supported a beneficial role of RBD trimer protein vaccination in reducing lung pathology, especially in the alum-3M-052 group. We also performed histopathologic examination of various other tissue samples including nasal turbinates, trachea, tonsils, hilar lymph nodes, spleen, heart, brain, gastrointestinal tract (stomach, jejunum, ileum, colon, and rectum), and testes. We did not observe significant histological lesions in upper respiratory tract tissues (nasal turbinates, trachea) and other examined tissues in control and vaccinated animals.

To understand the anamnestic expansion of antibody and T cells post-infection, we measured binding antibodies to RBD in serum on Days 0, 4, 7, and 10; and T cells in the blood at Day 10, following virus challenge. Five out of the 7 control animals became seropositive (RBD binding titer >2 fold compared to

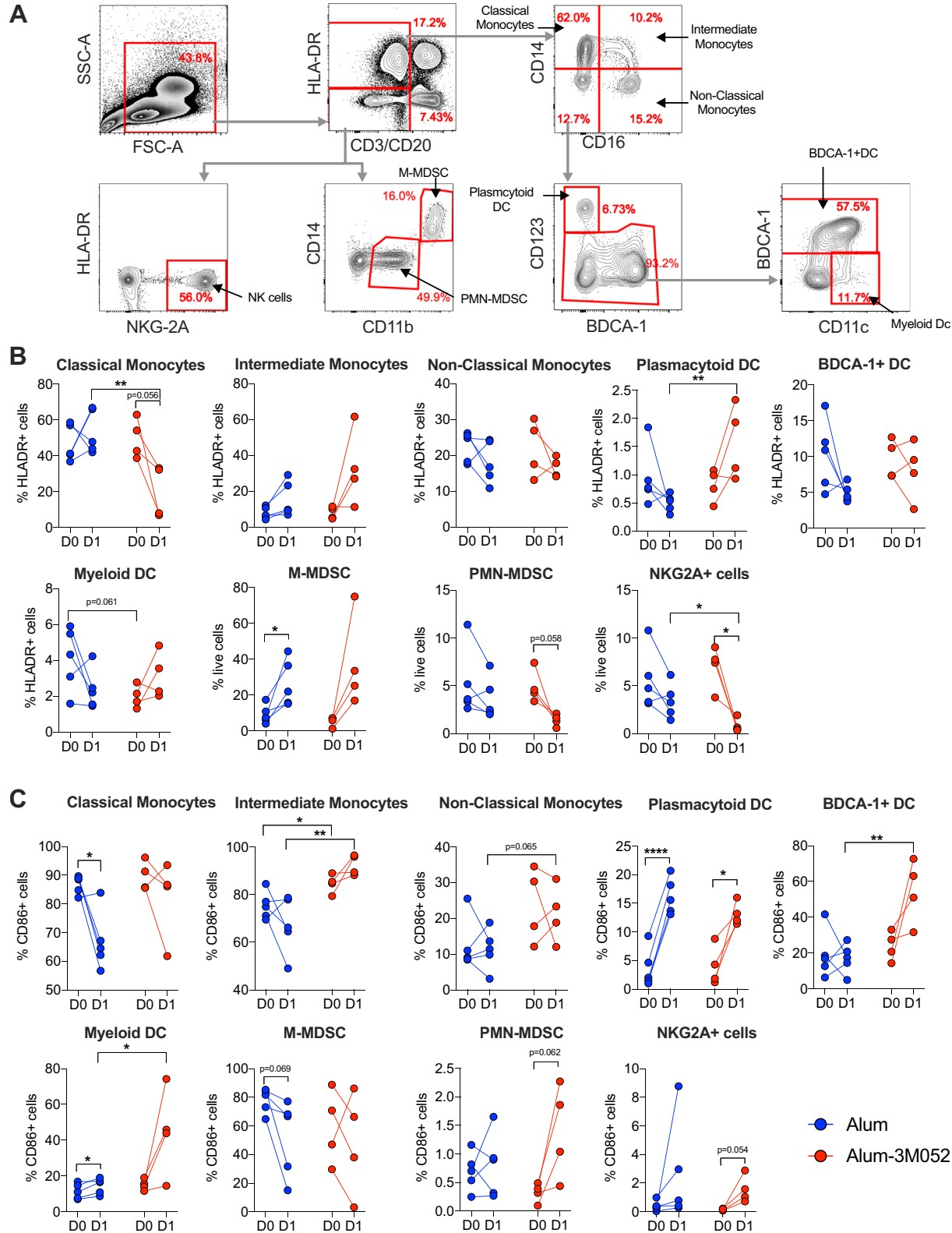

pre-infection) by Day 10 with a geometric mean titer of 651 that was 200-fold lower compared to the titer in vaccinated animals (Fig. 5H). In contrast, the RBD-binding antibody either stayed constant or decreased in all vaccinated animals except one animal each in alum and alum-3M-052 groups that showed a 2.4- and 7-fold increase compared to pre-infection titer (Fig. 5H). These were the same two animals that were positive for virus in the BAL

on Day 2 and in nasopharynx on Day 7 suggesting an anamnestic expansion of the antibody response following infection. Consistent with high virus replication, the control animals developed IFNg+ CD4 and CD8 T cell responses at Day 10, and most of the response was targeted against S1 and nucleocapsid (NC) proteins (Fig. 5I). Consistent with better protection and viral control, the T cell response in vaccinated animals was either low or negative and

**Fig. 4 Alum-3M-052 induces better innate activation compared to Alum in macaques. A** Flow cytometry gating strategy used to identify various innate cells in blood. Live cells were selected using live/dead marker and CD3+ and CD20+ cells were excluded. Then, different innate cells were defined using the following combination of markers. **B** and **C** Frequencies of various innate cells (**B**), Monocytes (HLA-DR+)—classical (CD14+), intermediate (CD14+ and CD16+) and non-classical (CD16+); PDCs (HLADR+ CD14− CD16− CD123+ BCDA1−); BDCA1+ DC (HLADR+ CD14− CD16− CD123− BCDA1+); MDCs (HLADR+ CD14− CD16− CD123− CD11c+); M-MDSC cells (HLADR− CD14+ CD11b+); PMN-MDSC cells (HLADR− CD14− CD11b+) and NK cells (HLADR− NKG2A+); and CD86+ activation (**C**). NK natural killer cells, PDC plasmacytoid dendritic cells, MDC myeloid-derived dendritic cells, myeloid-derived suppressor cells (MDSC), polymorphonuclear (PMN)—MDSC, and monocytic (M)—MDSC. Groups were color-coded; Alum: Blue, Alum-3M-052: Red. Bars and columns show mean responses in each group ± SEM; Statistical significance was measured using two-tailed paired *t*-test for comparisons within the group and two-tailed un-paired parametric *t*-test for comparisons between the groups. Asterisks denote statistical significance ∗*p* < 0.05 and ∗∗*p* < 0.01.

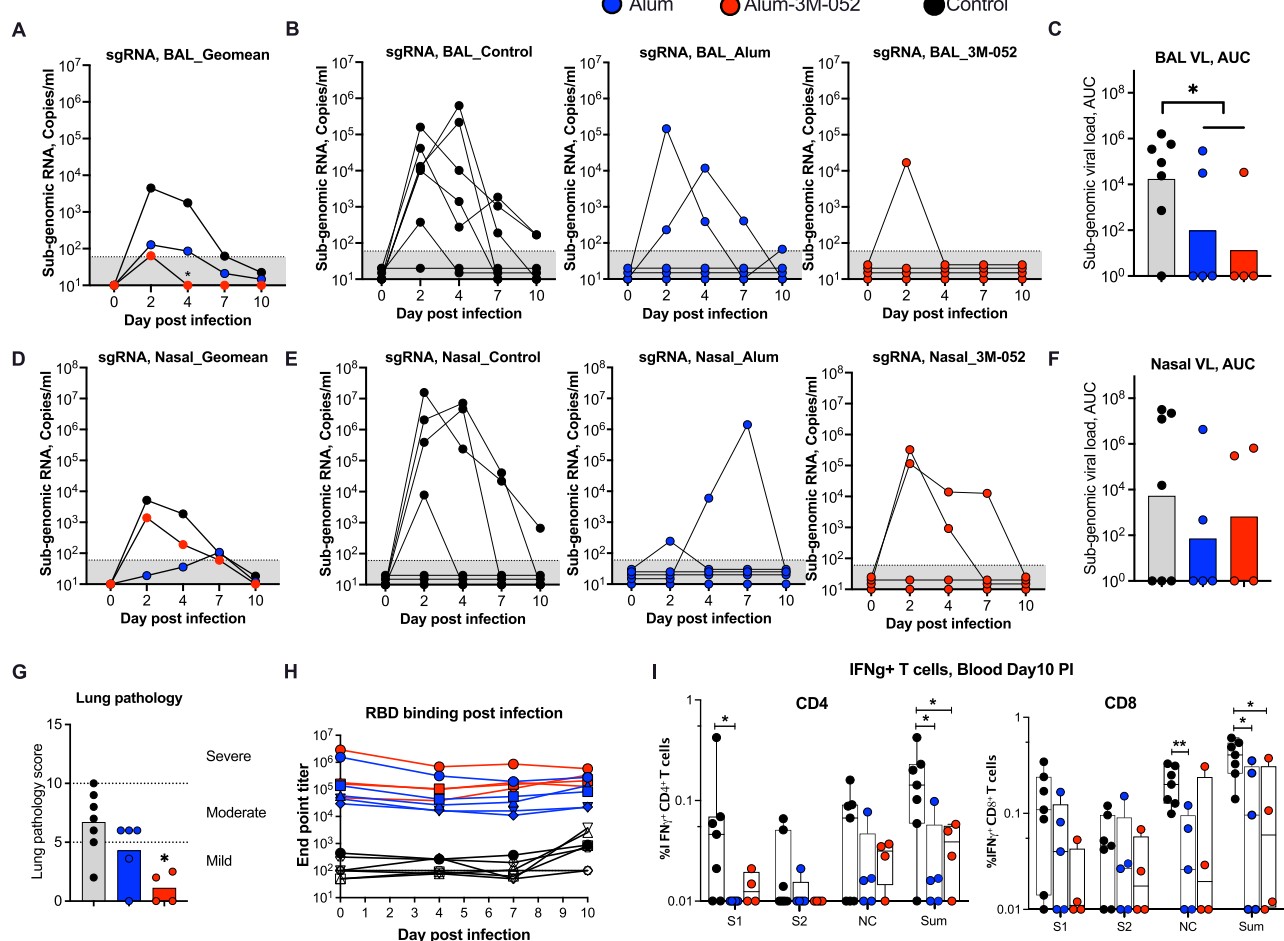

**Fig. 5 RBD trimer protein vaccination provides protection from SARS-CoV2 infection and replication in the lungs in macaques. A–F** SARS-CoV-2 subgenomic viral RNA copies/ml in the BAL fluids (Lungs) (**A–C**) and nasal fluids (**D–F**) on days 2, 4, 7, and 10 (day of euthanasia) post infection. Geomean for each group (**A, D**), data for individual animal (**B, E**) and the area under the curve (AUC) (**C, F**) are shown. The AUC was calculated based on viremia from Day 0 to Day 10. Comparison between non-vaccinated (*n* = 7) and vaccinated (*n* = 9) macaques was calculated using Student's *t*-test (unpaired, two-tailed). **G** Lung pathology score at Day 10 post-infection. Hematoxylin and Eosin-stained lung sections were used to analyze tissue structure and cell infiltration. The representative images for each animal presented in Fig. S2. See Table S1 for additional details. **H** SARS-CoV-2 RBD-specific serum IgG responses in serum following challenge. Each sample was analyzed in duplicates and repeated two independent times. **I** IFNγ + CD4+ and CD8+ T cells in blood at Day 10 post-infection after re-stimulation with peptide pool specific to indicated protein. S1, S1 region of spike; S2, S2 region of spike; NC, nucleocapsid; Sum, total response (S1 + S2 + NC). Bars and columns show mean responses in each group ± SEM; Mann–Whitney test: ∗*p* < 0.05, and ∗∗*p* < 0.001. Dotted lines reflect the limit of detection.

was significantly lower compared to controls (Fig. 5I). In conclusion, these results demonstrated that RBD trimer vaccination provides significant protection from infection and pathology in the lungs. Consistent with similar binding and neutralizing antibody response post-vaccination there was no significant difference between the two adjuvants at the dose tested in macaques.

**RBD trimer vaccination protects from infection-induced immune abnormalities in the lung**. To get further insights on the influence of infection and viral replication on innate immune cells, T follicular cells (Tfh), and B cells in the lung, we determined the frequency of various innate immune cells in BAL longitudinally, and the frequency of germinal center (GC) Tfh and GC-B cells in lung draining hilar lymph node on Day 10

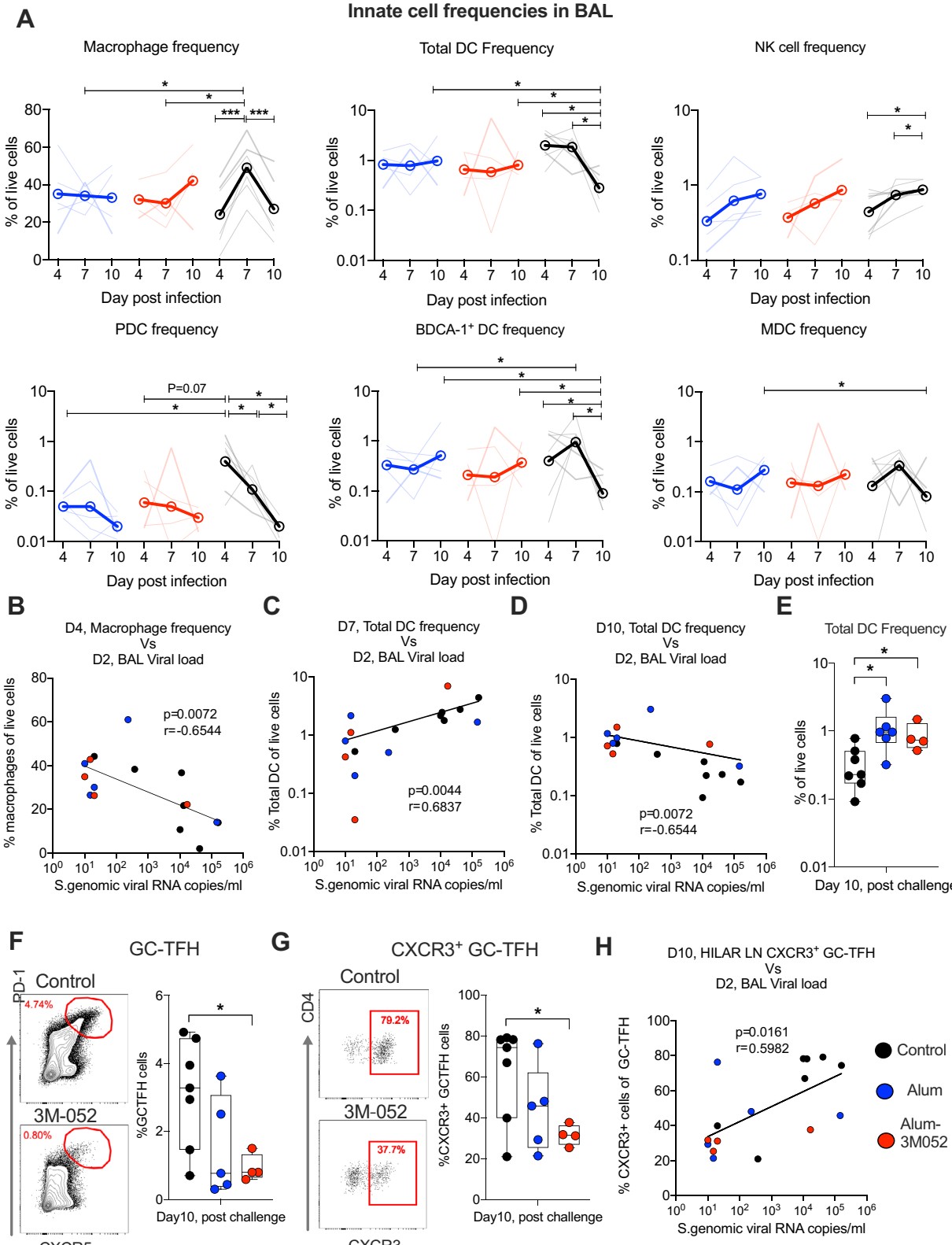

(Figs. 6 and S5). For innate cells we studied macrophages, three subsets of DC (PDC, BDCA-1+, and MDC), and NK cells on Days 4, 7, and 10. Among the total cells, the macrophages represented the majority with frequencies ranging from 10% to 70%, and the DC (HLA-DR+ cells without macrophages) and NK cells represented 0.1–1% (Fig. 6A). In controls, the frequency of macrophages increased from Day 4 to Day 7 and returned to Day 4 values by Day 10. However, the frequency of DCs declined sharply from Day 7 to Day 10, and the frequency of NK cells showed a gradual increase by Day 10. In vaccinated animals, however, the frequencies of all innate cells stayed relatively stable with occasional blips (Fig. 6A). In particular, the two vaccinated animals that showed clear virus replication in BAL showed an increase in the frequency of macrophages and DCs from Day 4 to

**Fig. 6 Immune responses in lungs and hilar lymph-node of rhesus macaques, post SARS-CoV-2 challenge. A** Following SARS-CoV-2 infection, BAL was collected on Days 4, 7, and 10, and frequencies of macrophages, total DCs, NK cells, PDCs, BDCA-1 DCs, and MDCs were measured. Gating strategy to identify innate cells in BAL fluid. Live cells were selected using live/dead marker and CD3$^+$ and CD20$^+$ cells were excluded. Then, different innate cells were defined using the following combination of markers: Macrophages (HLADR$^+$ CD163$^+$); PDCs (HLADR$^+$CD163$^-$CD123$^+$ BDCA1$^-$); BDCA1$^+$ DC (HLADR$^+$ CD163$^-$ CD123$^-$ BDCA1$^+$); MDCs (HLADR$^+$CD163$^-$CD123$^-$CD11C$^+$) and NK cells (HLADR$^-$ NKG2A$^+$). **B–D** Correlation analysis between Day 2 BAL viral loads (VL) and Day 4 macrophage frequencies (**B**); Day 2 BAL VL and Day 7 total DC frequencies (**C**); Day 2 BAL VL and Day 10 total DC frequencies (**D**). **E** Total DC frequencies in BAL fluid at Day 10 post-challenge in control, alum (Blue), and alum-3M-052 (Red) groups. **F** and **G** On Day 10 (at euthanasia), GC-Tfh (**F**) and CXCR3 GC-Tfh (**G**) cells were analyzed in hilar LN. Left: Representative flow plots; Right: Summary of data for all animals. CD3$^+$CD4$^+$CD8$^-$CXCR5$^{high}$PD-1$^{high}$ cells defined as GC-Tfh. **H** Correlation between Day 2 BAL VL and Day 10 HILAR LN CXCR3 GC-Tfh. The Spearman rank test was used to perform correlation analysis. NK, Natural killer cells. PDC, Plasmacytoid dendritic cells. MDC myeloid-derived dendritic cells, LN lymph node. Statistical differences between the groups and within the groups were analyzed using unpaired parametric *t*-test and paired-parametric *t*-test respectively. ∗*p* < 0.05, and ∗∗*p* < 0.001.

Day 7 and a decline from Day 7 to Day 10. The control animals showed significantly higher frequencies of PDC on Day 4 and macrophages on Day 7 compared to vaccinated animals. Correlations with Day 2 viral load revealed a negative association with macrophages on Day 4 (Fig. 6B) and a positive association with DCs on Day 7 (Fig. 6C). Consistent with a sharp decline in DCs from Day 7 to Day 10 in controls, the DC frequencies on Day 10 showed a negative association with Day 2 viral load (Fig. 6D). These results suggested that early (Day 2) high virus replication in the lungs may lead to early depletion of macrophages but the expansion of DCs, in particular PDC. Importantly, they show a gradual loss of DCs by Day 10 (Fig. 6E) despite most animals being negative for viruses demonstrating severe innate immune perturbations in the absence of vaccination. They also demonstrated that the vaccinated animals do not show significant changes for various innate immune cells consistent with better protection and viral control in these animals.

In the hilar lymph node at Day 10, we observed higher frequency of GC-TFH in the controls compared to the Alum-3M-052 group (Fig. 6F). However, the frequency of GC-Tfh was not significantly different between the control and Alum group. Similarly, the frequency of CXCR3 expression on the GC-Tfh (an indicator of pro-inflammatory environment) was also higher in the controls compared to Alum-3M-052 group but not the Alum group (Fig. 6F) and showed a direct correlation with Day 2 viral load in BAL (Fig. 6H). These observations suggested high viral antigen present in lymph node of control animals than vaccinated animals. Overall, these data indicated that vaccination lowers viral replication in the lungs and protects lungs from infection-induced pathology.

## Discussion

Effective vaccines that induce long-lived protective immunity are urgently needed to control the SARS-CoV-2 spread and the current pandemic[31,32]. Here, we designed and developed an RBD trimer protein and tested its immunogenicity and efficacy using two adjuvants in mice and macaques. Our results showed that RBD trimer induces a potent neutralizing antibody response with multiple antibody effector functions such as ADCD, ADCC, and ADCP, and provides protection against the SRAS-CoV-2 challenge in mice and macaques. They also showed that RBD trimer vaccination protects from lung pathology induced by SARS-CoV-2 infection. RBD trimer vaccinations induced Th1-biased CD4 T cell response which may be important for control of SARS-CoV-2 in humans[33,34]. As expected with protein-based vaccines, we did not observe the induction of CD8 T cell responses. As with most previous COVID-19 vaccine studies in macaques, the protection from infection and virus replication was mainly evident in the lower respiratory tract (lungs) (Fig. 5A–C) and not in the upper respiratory tract (nasopharynx) (Fig. 5D–F)[1,35,36]. However, the viral replication is more variable in the nasopharynx and therefore larger group sizes are required to make definitive conclusions

about vaccine efficacy on virus control in this compartment. Nevertheless, the neutralization titer, antibody effector functions, and protection we observed in this study are strong and merit further testing of RBD trimer as a potential vaccine immunogen for SARS-CoV-2.

Our results revealed that RBD trimer elicits higher neutralizing antibody responses compared to RBD monomer at 1 μg dose, which is 10-times lower than what is normally used in mice. It was interesting to note that the binding titer was comparable between the two immunogens and yet the neutralization titer was significantly higher in the RBD trimer group. We think the superior immunogenicity of RBD trimer can be attributed to the multimeric nature of the protein that resulted in higher activation and selection of high-affinity B cells. It is also possible that the specificity of neutralizing antibodies induced by the RBD trimer may be different from RBD monomer. Future studies will investigate these aspects and will determine the cross-reactivity of the induced antibody against newly emerging variants of SARS-CoV-2. Along the same lines, recent studies using dimeric[37] or multimeric display[38] of RBD protein have shown superior immunogenicity compared to RBD monomer. It is also important to note that the ability to use lower doses of protein will have important implications for human translation potential.

Multiple studies have used RBD-based immunogens as a vaccine for COVID-19 and tested for immunogenicity in animal models[15,19,37,39–41]. Most of these studies used RBD monomers as an immunogen, one study each used RBD dimers[37] or RBD 60mer displayed on a nanoparticle[19] as immunogen. Among these only one study investigated the protection against SARS-CoV-2 challenge in rhesus macaques[15]. While it will be informative to compare the immunogenicity and protection that we observed in this study with other RBD and non-RBD-based vaccine candidates, we think one needs to be careful since different studies used different neutralization assays that can provide different results. For example, one study reported that the neutralization titer measured using pseudovirus assay is nearly 10-times higher than the titer measured using the live virus neutralization[42]. In addition, some studies reported 50% neutralization titer and others 80–90%. Similarly, for protection studies, different stocks and doses of viruses, and species of macaques have been used. Importantly, most non-human primate studies used small group sizes (5–7 animals/group) and variability between each animal for neutralization titer can make comparisons harder. However, the emerging theme is that multimeric immunogens are superior to monomeric immunogens in inducing neutralizing antibody response against SARS-CoV-2.

Our mice studies clearly demonstrated that alum-3M-052 is superior to alum in inducing neutralizing antibody responses and protection. In mice we used a dose of 1 μg based on the previous titration in mice with another antigen. However, in rhesus macaques, we did not observe a significant difference between

alum and alum-3M-052 although there was a trend for better immune and protection outcomes with alum-3M-052. The non-significant differences between 3M-052 Alum and Alum alone could be due to an ineffective dose of 3M-52 in monkeys or the small sample size of animals being used. We think it is important to determine the optimal dose of alum-3M-052 for non-human primates and humans. The 10 μg dose of alum-3M-052 we used in our rhesus macaque study is 7.5 times lower than the dose we used for nanoparticle (NP) encapsulated 3M-052 (70 μg, NP-3M-052), which showed induction of high titer and persisting anti-body response at least for one year against HIV-1 envelope protein and provided protection in rhesus macaques[12–14].

In addition, NP-3M-052 induced the generation of plasma cells in bone marrow that are critical for the generation of persisting antibody titers. We did not measure plasma cells in bone marrow in the current study. However, given the activation of DCs and monocytes following vaccination in the alum-3M-052 adjuvanted group we predict that vaccination would induce plasma cells in the bone marrow and that will lead to the generation of long-lived neutralizing antibody response against SARS-CoV-2. Considering the protection induced by alum-3M-052 both in mice and macaques, our study supports further testing of alum-3M-052 as a potential adjuvant for SARS-CoV-2 protein-based vaccines.

We monitored the frequency of innate cells in BAL during the course of infection to understand the influence of virus replication in unvaccinated controls and how these changes relate to protection in vaccinated animals. Recent studies on SARS-CoV-2 infected rhesus macaques in unvaccinated animals[43,44] showed significant changes in frequencies of macrophages after infection. These studies also showed an increase in NKG2A+ NK cells[43,44] and PDC[43,44] post-infection. Consistent with these observations, in our study, we saw an increase in the frequency of NKG2A+ cells and higher PDCs during the early phase of infection (day 4 post-infection) in control animals, while vaccinated animals were relatively stable. Similar observations of decreased macrophages and increased PDCs in BAL were also reported during human infection[45]. Overall, these observations in BAL suggested relatively low infection and virus replication in vaccinated animals compared to control animals.

In conclusion, we demonstrated the potential benefits of RBD trimer as an immunogen for COVID-19 vaccine using mice and macaque models. Future studies will investigate the durability of the vaccine-induced immune responses and what parameters lead to protection.

## Methods

**Cells and viruses**. Human embryonic kidney (HEK)-293T cells, and Vero cells were obtained from ATCC. SARS-CoV-2 (icSARS-CoV-2) virus was obtained from BEI resources and grown in Suthar's laboratory at the Emory University. mNeonGreen SARS-CoV-2 (2019-nCoV/USA_WA1/2020) virus was produced by Pei Yong Shi's laboratory at the University of Texas[46]. The infectious clone SARS-CoV-2 (icSARS-CoV-2) was propagated in VeroE6 cells (ATCC) and sequenced[46]. The titer of SARS-CoV-2 viruses (icSARS-CoV-2 and 2019-nCoV/USA_WA1/2020) using VeroE6 cells. VeroE6 cells and HEK-293T cells were cultured in complete DMEM medium consisting of 1× DMEM (Corning, Cellgro), 10% fetal bovine serum (FBS), 25 mM HEPES buffer (Corning Cellgro), 2 mM L-glutamine, 1 mM sodium pyruvate, 1× non-essential amino acids, and 1× antibiotics. Viral stocks were stored at −80 °C until further use.

**Animal models**. Specific-pathogen-free (SPF) 6–8-week-old female BALB/c mice (00065 strain) were obtained from Jackson Laboratories (Wilmington, MA, USA) and housed in the animal facility at the Yerkes National Primate Research Center of Emory University, Atlanta, GA. Male Indian rhesus macaques (Macaca mulatta), 3–4.5 years old, were housed in pairs in standard non-human primate cages and provided with both standard primate feed (Purina monkey chow) fresh fruit, and enrichment daily, as well free access to water. Immunizations, blood draws, and other sample collections were performed under anesthesia with ketamine (5–10 mg/kg) or telazol (3–5 mg/kg) performed by trained research and veterinary staff.

**Construction and characterization of DNA/RBD trimer plasmid**. The plasmid containing amino acids residues 319–541 of SARS-CoV-2 spike protein fused to His tag was obtained from BEI resources (Cat# NR-52309). The plasmid expressing RBD monomer (DNA/RBD mono) was generated by cloning insert in pGA8 vector between NheI and AvrII restriction enzyme sites. Further, the plasmid expressing RBD trimer proteins (DNA/RBD tri) was generated by placing T4 trimerization sequence (GYIPEAPRDGQAYVRKDGEWVLLSTFL) at the C-terminus without His tag using In-Fusion cloning technology (Takara). Both plasmids contain the TPA signal sequence under the control of CMV promoter with intron A.

**Flow staining for RBD trimer protein expression**. HEK293 cells were transfected with 1 μg of plasmid (DNA/RBD tri) expressing trimerized from SARS-CoV-2 RBD. Cells were harvested, initially stained with live dead marker followed by cells were then fixed with Cytofix/cytoperm (BD Pharmingen), permeabilized with permwash (BD Pharmingen), and intracellularly stained with anti-SARS-CoV-2 RBD antibody (40592-T62, SinoBiological). Later, donkey anti-rabbit IgG coupled to PE (406421, BioLegend) secondary antibody was used to confirm the expression.

**Protein expression and purification**. Monomeric and trimeric form of RBD proteins of SARS-CoV-2 was produced by transfecting FreeStyle 293-F cells using plasmids DNA/RBD mono and DNA/RBD tri, respectively. Transfections were performed according to manufacturer's instructions (Thermo Fisher). Briefly, FreeStyle 293-F cells were seeded at a density of $2 \times 10^6$ cells/ml in Expi293 expression medium and incubated in an orbital shaking incubator at 37 °C and 127 rpm with 8% $CO_2$ overnight. Next day, $2.5 \times 10^6$ cells/ml were transfected using ExpiFectamine$^{TM}$ 293 transfection reagent (ThermoFisher, cat. no. A14524) as per manufacturer's protocol. The cells were grown for 72 h at 37 °C,127 rpm, 8% $CO_2$. The cells were removed by centrifugation at 2000×g for 10 minutes at 4 °C, the supernatant was collected and filtered using a 0.22 μm stericup filter (Thermo-Fisher, cat. no. 290-4520) and loaded onto pre-equilibrated affinity column for protein purification. The SARS-CoV-2 RBD monomer (with His tag) and RBD trimer proteins were purified using Ni-NTA resin (ThermoFisher, cat. no. 88221) and Agarose bound conA (Vector Labs, cat. no. AL-1243-5) respectively. Briefly, His-Pur Ni-NTA resin was washed with PBS by centrifugation at 2000×g for 10 min. The resin was resuspended with the supernatant and incubated for 2 h on a shaker at RT. Polypropylene column was loaded on the supernatant–resin mixture and washed with wash buffer (25 mM Imidazole, 6.7 mM $NaH_2PO_4$·$H_2O$, and 300 mM NaCl in PBS) four times, after which the protein was eluted in elution buffer (235 mM Imidazole, 6.7 mM $NaH_2PO_4$·$H_2O$ and 300 mM NaCl in PBS). RBD trimer protein supernatants were mixed with ConA Agarose-resin overnight on rocker at 4 °C. The supernatant-resin mix was loaded onto the column and washed three times with PBS and eluted using 1 M methyl-α-D mannopyranoside (pH 7.4). Eluted proteins were dialyzed against PBS using Slide-A-lyzer Dialysis Cassette (ThermoScientific, Cat# 66030) and concentrated using either 10 kDa Amicon Centrifugal Filter Units (for RBD-mono) or 50 kDa Amicon Centrifugal Filter Units (for RBD-tri) at 2000×g at 4 °C. The concentrated protein elutes were run on a Superdex 200 Increase 10/300 GL (GE Healthcare) column on an Akta$^{TM}$Pure (GE Healthcare) system and collected the peak that is matching to corresponding protein. The quantity of the proteins was estimated by BCA Protein Assay Kit (Pierce) and quality by BN-PAGE (NuPAGE™, 4–12% Bis–Tris Protein Gels, ThermoScientific), SDS–PAGE, and Western blot. Image Lab 5.2 version software was used to acquire images.

**Western blotting**. HEK293-T cells were transfected with 1 μg of plasmid (DNA/RBD-monomer and DNA/RBD-trimer) expressing monomer and trimerized forms of SARS-CoV-2 RBD. Cells were harvested and lysed in ice-cold RIPA buffer and supernatants were collected. Lysates were kept on ice for 10 min, centrifuged, and resolved by SDS–PAGE using precast 4–15% SDS polyacrylamide gels (BioRad). Proteins were transferred to a nitrocellulose membrane, blocked with 1% casein blocker overnight (Cat# 1610782 BioRad), and incubated for 2 h at room temperature with primary RBD antibody (Cat# 40592-T62, Sino Biologicals) diluted 1:4000 in blocking buffer. The membranes were washed in PBS containing Tween-20 (0.05%) and incubated for 1 h with horseradish peroxidase-conjugated anti-rabbit secondary antibody (Cat #4030-05, Southern Biotech). The membranes were washed, and proteins were visualized using the ECL select chemiluminescence substrate (Cat# RPN2235, GE Healthcare).

**Animal vaccination**. BALB/c mice of 6–8-week-old female were immunized with 1–10 μg SARS-CoV-2 RBD trimer or monomer protein, intramuscularly, at weeks 0 and 4 with either aluminum hydroxide (100 μg) or Alum- 3M-052 (100 μg alum, 1 μg 3M-052). Antigen was mixed with the adjuvant and incubated at 4 °C for 15 min on a rocker before immunization. The blood samples were collected at two weeks following each immunization by facial vein puncture in BD Microtainer® Tube for analyzing SARS-CoV-2 RBD-specific serum antibody responses.

*SARS-CoV-2 (MA10) challenge*. All SARS-CoV-2 (MA10) challenge experiments were carried out at the University of North Carolina (UNC) at Chapel Hill. Briefly, mice were anesthetized using ketamine/xylazine and infected intranasally with $10^5$ PFU SARS-CoV-2 MA10 strain diluted in PBS[28]. Briefly, the mouse-adapted SARS-

CoV-2 virus (MA10) was generated from an infectious clone of SARS-CoV-2 MA stock and further genetically engineered to introduce Q498Y/P499T substitutions into the spike protein. SARS-CoV-2 MA10 stock was generated from a P10-infected mouse lung homogenate via inoculation of Vero E6 cells. Clinical signs of disease (weight loss and body score) were monitored daily. The mice were euthanized by isoflurane overdose at indicated time points when samples for titer (caudal right lung lobe) were collected. Plaque assay was used to define Lung viral titers. Briefly, right caudal lung lobes were homogenized in 1 ml PBS using glass beads and serial dilutions of the clarified lung homogenates were added to a monolayer of Vero E6 cells. After 3 days plaques were visualized via staining with Neutral Red dye and counted.

*Rhesus macaque study.* A total of 16 Indian origin rhesus macaques (Macaca mulatta; male) from 4 to 6 years of age were included in this study. The animal study was conducted at Yerkes National Primate Research Center, Emory University, and was approved by the Emory IACUC. Nine macaques were randomly allocated into two vaccine groups. Group 1 ($n = 5$) received 30 μg RBD trimer antigen adjuvanted with 1 mg amounts of Aluminum Hydroxide. Group 2 ($n = 4$) received 30 μg RBD trimer antigen adjuvanted with Alum (1 mg)-3M-052 (10 μg) formulations. Macaques were immunized intramuscularly with indicated RBD trimer-adjuvanted vaccines, at week 0 and 4, respectively with a total volume of 1 ml in the thigh. The serum samples and PBMCs were collected at the indicated time points and subjected to immunological assays. The remaining 7 animals were used as controls. Of the 7, 5 animals received wild-type modified vaccinia Ankara (MVA) vaccine as reported previously[47] and 2 did not receive any vaccine.

*SARS-CoV-2 challenge.* 4 weeks after final immunization the macaques were challenged with a total of $5 \times 10^4$ pfu ($2.5 \times 10^4$ pfu/ml) of SARS-CoV-2 (2019-nCoV/USA_WA1/2020). Virus was administered as 1 ml by IT and 1 ml by intranasal (IN) route (0.5 ml in each nostril). Nasal swabs and BAL samples were collected, stored immediately in the viral transport media, and processed for viral RNA extraction on the same day. Starting from the day of challenge, the nasal swabs and BAL fluid were collected on day 2, 4, 7, and 10 and subjected to viral load measurements. At Day 10 after SARS-CoV-2 challenge, all vaccinated and non-vaccinated macaques were euthanized. Necropsy samples were collected (lung tissues) and were subjected to Hematoxylin and Eosin staining.

**Binding antibody responses using ELISA.** SARS-CoV-2 S (RBD, S1, and S)-specific IgG in serum and BAL was quantified by enzyme-linked immunosorbent assay (ELISA)[48]. Briefly, Nunc high-binding ELISA plates were coated with 2 μg/ml of recombinant SARS-CoV-2 proteins (RBD, S1, and S) proteins in Dulbecco's phosphate-buffered saline (DPBS) and incubated overnight at 4 °C. SARS-CoV-2 RBD and S1 proteins were produced in the lab whereas, S1 and S (S1 + S2 ECD) proteins were purchased. Plates were then blocked with 5% blotting-grade milk powder and 4% whey powder in DPBS with 0.05% Tween 20 for 2 h at room temperature (RT). Plates were then incubated with serially diluted serum samples (starting from 100, 3-fold, 8x) and incubated for 2 h at RT followed by six washes. Total SARS-CoV-2 S (RBD, S1, and S)-specific mouse IgG and monkey IgG antibodies were detected using HRP-conjugated anti-mouse (1:6000) (Southern Biotech; AL, USA) and goat anti-monkey IgG secondary antibody (1:10,000), respectively for 1 h at RT. The plates were washed and developed using TMB (2-Component Microwell Peroxidase Substrate Kit) and the reaction was stopped using 1 N phosphoric acid solution. Plates were read at 450 nm wavelength within 30 min using a plate reader (Molecular Devices, San Jose, CA, USA). ELISA endpoint titers were defined as the highest reciprocal serum dilution that yielded an absorbance >2-fold over background values.

**Live-virus neutralization.** Live-virus SARS-CoV-2 neutralization antibodies were assessed using a full-length mNeonGreen SARS-CoV-2 (2019-nCoV/USA_WA1/2020), generated as previously described[46]. Vaccinated mice, NHP, and post-challenge sera were incubated at 56 °C for 30 min and manually diluted in duplicate in serum-free Dulbecco's modified Eagle medium (DMEM) and incubated with 750–1000 focus-forming units (FFU) of infectious clone-derived SARS-CoV-2-mNG virus[46] at 37 °C for 1 h. The virus/serum mixture was added to VeroE6 cell (C1008, ATCC, #CRL-1586) monolayers, seeded in 96-well blackout plates, and incubated at 37 °C for 1 h. Post incubation, the inoculum was removed and replaced with pre-warmed complete DMEM containing 0.85% methylcellulose. Plates were incubated at 37 °C for 24 h. After 24 h, the methylcellulose overlay was removed, cells were washed three times with phosphate-buffered saline (PBS), and fixed with 2% paraformaldehyde (PFA) in PBS for 30 min at room temperature. PFA is then removed and washed twice with PBS. The foci were visualized using an ELISPOT reader (CTL ImmunoSpot S6 Universal Analyzer) under a FITC channel and enumerated using Viridot[49]. Viridot has only one version but, was written to work with versions up to R 3.4.1. This was mentioned in the details. The neutralization titers were calculated as follows: 1—ratio of the (mean number of foci in the presence of sera: foci at the highest dilution of respective sera sample). Each specimen is tested in two independent assays performed at different times. The focus-reduction neutralization mNeonGreen live-virus 50% titers (FRNT-mNG$_{50}$) were interpolated using a 4-parameter nonlinear regression in GraphPad Prism 8.4.3. Samples that did not neutralize at the limit of detection at 50% were plotted at 10 and were used for geometric mean calculations.

**Antibody isotype, IgG subclass, and FcR binding of mouse sera.** For relative quantification of antigen-specific antibody titers, a customized multiplexed approach was applied, as previously described[50]. Therefore, magnetic microspheres with a unique fluorescent signature (Luminex) were coupled with SARS-CoV-2 antigens including spike protein (S) (provided by Eric Fischer, Dana Farber), RBD, and CoV HKU1 RBD (provided by Aaron Schmidt, Ragon Institute), CoV-2 S1 and S2 (Sino Biologicals) as well as influenza as control (Immune Tech). Coupling was performed using EDC (Thermo Scientific) and Sulfo-NHS (Thermo Scientific) to covalently couple antigens to the beads. $1.2 \times 10^3$ beads per region/ antigen were added to a 384-well plate (Greiner) and incubated with diluted plasma samples (1:90 for all readouts) for 16 h while shaking at 900 rpm at 4 °C, to facilitate immune complex formation. The next day, immune complexed microspheres were washed three times in 0.1% BSA and 0.05% Tween-20 using an automated magnetic plate washer (Tecan). Anti-mouse IgG-, IgG2a-, IgG3-, IgA- and IgM-PE coupled (Southern Biotech) detection antibodies were diluted in Luminex assay buffer to 0.65 μg/ml. Beads and detection antibodies were incubated for 1 h at RT while shaking at 900 rpm. Following washing of stained immune complexes, a tertiary goat anti-mouse IgG-PE antibody (Southern Biotech) was added and incubated for 1 h at RT on a shaker. To assess Fc-receptor binding, mouse Fc-receptor FcγR2, FcγR3, FcγR4 (Duke Protein Production facility) were biotinylated (Thermo Scientific) and conjugated to Streptavidin-PE for 10 min (Southern Biotech) before adding to immune complexes and processed as described above. Finally, beads were washed and acquired on a flow cytometer, iQue (Intellicyt) with a robot arm (PAA). Events were gated on each bead region, median fluorescence of PE for bead positive events were reported. Samples were run in duplicate for each secondary detection agent.

**Antibody isotype, IgG subclass, and FcR binding of monkey sera.** A Luminex assay was used to detect and quantify antigen-specific subclass, isotype, and Fc-receptor (binding) factors[50]. With this assay, we measured the antibody concentration against SARS-CoV-2 RBD (kindly provided by Aaron Schmidt, Ragon Institute) and SARS-CoV-2 S (kindly provided by Erica Ollmann Saphire, La Jolla Institute). Carboxylate-modified microspheres (Luminex) were activated using EDC and Sulfo-NHS and antigens were covalently bound to the beads via NHS-ester linkages. Antigen-coupled beads were washed and blocked. Immune complexes were formed by mixing appropriately diluted plasma (1:100 for IgG1, IgG2, IgG3, IgG4, IgA, IgM, and 1:1000 for FcγRs) to antigen-coupled beads and incubating the complexes overnight at 4 °C. Immune complexes were then washed with 0.1% BSA 0.02% Tween-20. PE-coupled secondary antibodies for each antibody isotype or subclass (Southern Biotech) were used to detect antigen-specific antibody titer. For FcRs, biotinylated FcRs were labeled with streptavidin-PE before addition to immune complexes. Fluorescence was measured with an iQue (Intellicyt) and analyzed using Forecyt (v 8.1) software. Data are reported as median fluorescence intensity (MFI).

**ADCP, ADNP, and ADCD assays for monkey sera.** ADCP, ADNP, and ADCD were measured as previously described[51–53]. For ADCP and ADNP, yellow-green fluorescent neutravidin beads were coupled to biotinylated SARS-CoV-2 S or RBD. For ADCD, red fluorescent neutravidin beads were coupled to biotinylated SARS-CoV-2 S or RBD. Antigen-coupled beads were then incubated with appropriately diluted plasma (ADCP 1:100, ADNP 1:50, ADCD 1:10) for 2 h at 37 °C to form immune complexes. For ADCP, THP-1s (ATCC) were added at $1.25 \times 10^5$ cells/ml and incubated for 16 h at 37 °C. For ADNP, leukocytes were isolated from fresh peripheral whole blood by lysing erythrocytes using ammonium-chloride potassium lysis. Leukocytes were added to immune complexes at $2.5 \times 10^5$ cells/ml and incubated for 1 h at 37 °C. Neutrophils were detected using anti-human CD66b Pacblue (Biolegend). For ADCD, lyophilized guinea pig complement (Cedarlane) was resuspended, diluted in gelatin veronal buffer with calcium and magnesium (GVB++, Boston BioProducts), and added to immune complexes. The deposition of C3 was detected using an anti-C3 FITC antibody (Mpbio).

All functional assays were acquired with an iQue (Inellicyt) and analyzed using Forecyt software. For ADCP, events were gated on singlets and fluorescent cells. For ADNP, bead-positive neutrophils were defined as CD66b positive, fluorescent cells. For both ADCP and ADNP, a phagocytic score was defined as (percentage of bead-positive cells) × (MFI of bead-positive cells) divided by 10,000. For ADCD, data were reported as median fluorescence of C3 deposition (MFI).

**Cell processing.** For mouse studies, spleens and lungs of vaccinated and control animals were removed and placed on ice in cold RPMI 1640 (1×) with 5% FBS (Company, state, USA). 1× β-Mercaptoethanol (Invitrogen, State, USA) was added to the complete medium to isolate splenocytes. Whereas lungs were cut into small pieces and incubated at 37 °C in RPMI (1×) medium containing Collagenase type IV and DNase I with gentle shaking for 30 min. After incubation, cells were isolated by forcing tissue suspensions through a 70 μM cell strainer. RBCs were removed by ACK lysis buffer and live cells counted by trypan blue exclusion. For macaques, PBMC from blood collected in sodium citrate CPT tubes was isolated using standard procedures. Post SARS-CoV-2 challenge, samples were processed and stained in BSL-3 facility.

For BAL fluid processing and single-cell isolation, up to 50 ml physiological saline was delivered through the trachea into the lungs of anesthetized animals using a camera-enabled fiberoptic bronchoscope. The flushed saline was re-

aspirated five times before pulling out the bronchoscope. This collection was filtered through 70 μm cell strainer and centrifuged at 1126×g for 5 min. Pelleted cells were suspended in 1 ml R10 medium (RPMI(1×), 10% FBS) and stained as described in the sections below.

For processing lymph-node, lymph-node biopsies were dissociated using 70 μm cell strainer. The cell suspension was washed twice with R-10 media.

**Intracellular cytokine staining (ICS).** Functional responses of SARS-CoV-2 RBD, S1 and S2 specific CD8+ and CD4+ T cells in vaccinated animals were measured using peptide pools and intracellular cytokine staining (ICS) assay[54]. Overlapping peptides (13 or 17 mers overlapping by 10 amino acids) were obtained from BEI resources (NR-52402 for spike and NR-52419 for NC) and different pools (S1, S2, RBD, and NC) were made. The S1 pool contained peptides mixed from 1 to 97, S2 pool contained peptides mixed from 98 to 181, RBD pool contained peptides 46–76 and NC pool contained 57 peptides. Each peptide was used at 1 μg/ml concentration in the stimulation reaction. Two million cells suspended in 200 μl of RPMI 1640 medium with 10% FBS were stimulated with 1 μg/ml CD28 (BD Biosciences), 1 μg/ml CD49d (BD Biosciences) co-stimulatory antibodies, and different peptide pools. These stimulated cells were incubated at 37 °C in 5% $CO_2$-conditioned incubator. After 2 h of incubation, 1 μl Golgi-plug and 1 μl Golgi-stop/ml (both from BD Biosciences) were added and incubated for 4 more hours. After total 6 h of incubation, cells were transferred to 4 °C overnight and were stained the next day. Cells were washed once with FACS wash (1XPBS, 2% FBS, and 0.05% sodium azide) and surface stained with Live/Dead-APC-Cy7, anti-CD3, anti-CD4, and anti-CD8, each conjugated to a different fluorochrome for 30 min at RT. The stained cells were washed once with FACS wash and permeabilized with 200 μl of cytofix/cytoperm for 30 min at 4 °C. Cells were washed once with perm wash and incubated with anti-cytokine antibodies for 30 min at 4 °C. Finally, the samples were washed once with perm wash and once with FACS wash, and fixed in 4% paraformaldehyde solution for 20 min before acquiring on BD LSR Fortessa flow cytometer (v8.0.1). Data were analyzed using FlowJo software.

**Histopathological examination.** For histopathologic examination in macaques, the animals were euthanized due to the study endpoint, and a complete necropsy was performed. For histopathologic examination, various tissue samples including lung, nasal turbinates, trachea, tonsils, hilar lymph nodes, spleen, heart, brain, gastro-intestinal tract (stomach, jejunum, ileum, colon, and rectum), testes were fixed in 10% neutral-buffered formalin for 24 h at room temperature, routinely processed, paraffin-embedded, sectioned at 4 μm, and stained with hematoxylin and eosin (H&E). The H&E slides from all tissues were examined by two board-certified veterinary pathologists. For each animal, all the lung lobes were used for analysis, and affected microscopic fields were scored semiquantitatively as Grade 0 (None); Grade 1 (Mild); Grade 2 (Moderate); and Grade 3 (Severe). Scoring was performed based on these criteria: number of lung lobes affected, type 2 pneumocyte hyperplasia, alveolar septal thickening, fibrosis, perivascular cuffing, peribronchiolar hyperplasia, inflammatory infiltrates, hyaline membrane formation. An average lung lobe score was calculated by combining scores from each criterion. For animals with multiple affected lung lobes, each lung lobe was assessed individually and then the scores for each category were averaged. The total score was then determined for each animal. Digital images of H&E stained slides were captured at ×100 and ×200 magnification with an Olympus BX43 microscope equipped with a digital camera (DP27, Olympus) using Cellsens® Standard 2.3 digital imaging software (Olympus).

**Immunophenotyping of BAL and LN cells.** Briefly, the cells were stained with a surface antibody cocktail and incubated at RT for 30 min. The stained cells were given a FACS wash and permeabilized with 1 ml perm buffer (Invitrogen) for 30 min at RT. These cells were given a perm wash (Invitrogen) and stained with an intracellular antibody cocktail for 30 min at RT. Finally, the cells were washed once with perm wash and a FACS wash and fixed in 4% paraformaldehyde solution for 20 min before acquiring on BD LSR-II flow cytometer (v8.0.1). Samples prior to the challenge were acquired without 20-min 4% paraformaldehyde fixation.

BAL innate cell surface antibody cocktail: live/dead stain-APC-Cy7, anti-CD3-BV605, anti-CD20-BV605, anti-NKG2A-APC, anti-HLADR-PERCP, anti-CD11b-PE/Dazzle 594, anti-163-eFluor-450, anti-CD123-PE-Cy7, anti-CD11c-BV655, and anti-BDCA1-BV711. BAL innate cell intracellular antibody: anti-Ki67-BV786. T-cell phenotype surface antibody cocktail: live/dead stain-APC-Cy7, anti-CD3-PerCP, anti-CD4-BV655, anti-CD8-BV711, anti-PD1-BV421, anti-CXCR5-PE, and anti-CXCR3-BV605. T-cell phenotype intracellular antibody: anti-Ki67-BV786. B-Cell phenotype surface antibody cocktail: live/dead stain-APC-Cy7, anti-CD3-AF700, and anti-CD20-BV605, B-cell phenotype intracellular antibody: anti-BCL6-PE-CF594 and anti-Ki67-PE-Cy7.

**Viral RNA extraction and quantification.** SARS-CoV-2 genomic and subgenomic RNA was quantified in naso-pharyngeal (NP) swabs, and brocho-alveolar lavages (BAL). Swabs were placed in 1 ml of Viral Transport Medium (VTM; Labscoop (VR2019-1L)). Viral RNA was extracted from NP swabs, throat swabs, and BAL on fresh specimens using the QiaAmp Viral RNA mini kit according to the manufacturer's protocol. Quantitative PCR (qPCR) was performed on subgenomic using primer and probe align the subgenomic mRNA transcript of the E gene[55] are

SGMRNA-E-F: 5′-CGATCTCTTGTAGATCTGTTCTC-3′, SGMRNA-E-R: 5′-ATATTGCAGCAGTACGCACACA-3′, and SGMRNA-E-Pr: 5′-FAM-ACACTA GCCATCCTTACTGCGCTTCG-3′ (Table S2). qPCR reactions were performed in duplicate with the Thermo-Fisher 1-Step Fast virus master mix using the manufacturer's cycling conditions, 200 nM of each primer, and 125 nM of the probe. The limit of detection in this assay was about 128 copies per ml of VTM/BAL depending on the volume of extracted RNA available for each assay. To verify sample quality the CDC RNase P p30 subunit qPCR was modified to account for rhesus macaque specific polymorphisms. The primer and probe sequences are RM-RPP30-F 5′-AGACTTGGACGTGCGAGCG-3′, RM-RPP30-R 5′- GAGCCGCTGT CTCCACAAGT-3′, and RPP30-Pr 5′-FAM-TTCTGACCTGAAGGCTCTGCGC G-BHQ1-3′[56]. A single well from each extraction was run as described above to verify RNA integrity and sample quality via detectable and consistent cycle threshold values (Ct between 25 and 32).

**Quantification and statistical analysis.** GraphPad Prism version 8.4.3 (471) (GraphPad Software) was used to perform data analysis and statistics. The difference between any two groups at a time point was measured either using a two-tailed nonparametric Mann–Whitney rank-sum test or unpaired parametric t-test depending on the distribution of the data. Comparisons between different time points within a group used paired parametric t-test. P-value of <0.05 was considered significant. The correlation analysis was performed using the Spearman rank test.

**Reporting summary.** Further information on research design is available in the Nature Research Reporting Summary linked to this article.

## Data availability

All data supporting the experimental findings of this study are available within the manuscript and are available from the corresponding author upon request. Source data are provided with this paper.

## Materials availability

All unique/stable reagents generated in this study are available from the Lead Contact up on reasonable request after completion of a Materials Transfer Agreement. Source data are provided with this paper.

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

## Acknowledgements

We thank Traci Legere, Brenda Wehrle, and Zeba Momin for help with processing of blood and tissue samples, Shelly Wang for help with viral RNA mesurements, the Yerkes Division of Pathology and Research Resources for outstanding animal care during the pandemic and Histology and Molecular Pathology Lab for help with tissue sectioning. The following reagent was produced under HHSN272201400008C and obtained through BEI Resources, NIAID, NIH: Vector pCAGGS Containing the SARS-Related Coronavirus 2, Wuhan-Hu-1 Spike Glycoprotein Receptor Binding Domain (RBD), NR-52309. This work was supported in part by National Institutes of Health Grants RO1 AI148378-01S1 and Fast Grants Award #2166 to R.R.A., and NCRR/NIH base grant P51 OD011132 to theYerkes National Primate Research Center.

## Author contributions

R.R.A. designed the RBD trimer protein and responsible for overall experimental design and supervision of laboratory studies. N.C. constructed and characterized the RBD trimer protein vaccine. N.K.R., N.C., V.S.B., Sailaja Gangadhara, V.V.E., L.L., Anusmita Sahoo, Ayalensh Shiferaw, T.M.S., K.F., S.F., S.A.S. were responsible for conducting experiments, data collection and data analysis. Sanjeev Gumber and S.K. performed H&E staining and determined lung pathology scores. S.F., C.A., and S.A.S., performed antibody effector function assays under the supervision of G.A., T.H.V. supervised viral RNA measurements. M.T. and C.F. provided the alum-3M-052 adjuvant. P.-Y.S. and V.D.M. provided virus for neutralization assays and challenge studies. L.G. and K.H.D. performed challenge studies in mice. M.S.S. conducted and supervised live virus neutralizing antibody assays. R.R.A. and N. K.R. wrote the manuscript. All authors contributed to manuscript editing.

## Competing interests

R.R.A., N.C. and N.K.R are co-inventors of RBD trimer vaccine technology. Emory university filed a patent on this technology. C.B.F is an inventor on a patent application of the 3M-052-Alum formulation. All other authors declare no competing interests.

### Ethics declarations

Mice and rhesus macaques were housed at the Yerkes National Primate Research Center and animal experiments were approved by the Emory University Institutional Animal Care and Use Committee (IACUC) using protocols PROTO201700014 and PROTO202000057. All animal experiments were carried out in accordance to USDA regulations and recommendations derived from the Guide for the Care and Use of Laboratory Animals. Mice were shipped to the University of North Carolina (UNC) at Chapel Hill and performed SARS-CoV-2 (MA10) challenge experiments using IACUC protocol 20-114.
