## [Peer Review File · Nature Communications]

Reviewers' Comments:

Reviewer #1:

Remarks to the Author:

In this manuscript Routhu and colleagues describe the immunogenicity and efficacy of RBD vaccine as a monomer or a trimer in mice and test the trimer adjuvanted with alum or TLR7/8 agonist formulation alum -3M-052 in monkeys.

The trimeric adjuvanted vaccines protect animals from VL and disease and mice receiving the alum -3M-052 have higher ab titers including neutralization.

In monkeys, the 2 adjuvants are comparable in protection while differing in some instances after infection, for example in the frequency of innate cells. The authors comment that the possible lack of difference may be due to the dose or the sample size.

Overall, the draft needs careful revision for typos, repetitions, and for improving descriptions of the data in a scientific appropriate manner (examples: "quite strong" for the antibodies titers "strong protection";

In particular, protection needs to be defined.

The reason why the alum -M-052 was proposed is to study its potential in enhancing durability of responses, yet duration of such responses was not studied in this study.

It would be helpful to include analysis of B cells in bone marrow (LLPC), if available, or plasmablasts in lymph nodes to look for differences suggesting longevity of responses in alum vs. alum-M-052.

In discussion

There is a lack of contextualization in previously published studies on correlates in protection in humans and macaques (innate and adaptive) and particularly on similar vaccines tested in the same models.

In particular, discussion should include

1) A more inclusive comparison with similar vaccines (RBD immunogens) tested in same models with different results in macaques and mice (example: Tan, HX. Ref15);

2) Discussion about innate responses as correlates of risk in macaques and humans (Fahlberg MD Nat Commun. 2020 Nov 27;11(1):607, Singh, D.K, Nat Microbiol 6, 73–86)(Wilk, A. J. et al. A single-cell atlas of the peripheral immune response in patients with severe COVID-19. Nat. Med. 26, 1070–1076 (2020). Would help explaining the analysis and findings in Figure 4.

Moreover

-“They also showed that RBD trimer vaccination protects from lung pathology induced by vaccination”. not clear:

-“Vaccinations induced Th1-biased CD4 T cell response which is important for protection against SARS-CoV-2”. Reference needed

-“The protection was mainly evident in the lower respiratory tract (lungs) and to a lesser extent in the upper respiratory tract (nasopharynx)”. Define protection and figures when this is shown.

-“The viral load is more variable in the nasopharynx and larger group sizes are required to make definitive conclusions about vaccine efficacy in this compartment” contradicts what just stated, not clear.

Results

Fig 2D Is the correlation significant when the 2 groups are analyzed separately? It appears that the difference between the 2 groups included is what is driving it.

Methods:

SARS COV2 challenge of macaques, which virus?

Figures

-Figure 2 should include the timing under figures (prime and boost). What is the statistical analysis for

B, C, E, F, G, H I is it corrected for multiple comparison? Why there is only 3 animals for the viral load Fig 2I? There are N= 5 animals described yet there are 6 animals per group in Figure 2D. Figure 2F: what is the significance here? With the control?

-Figure 3: Are the stars representative of significance with controls? Are these corrected? What is the color of the legend corresponding to in F? and it is that Person test correlaplot?

FIGURE S1: 3 groups presented. Statistical differences with what? FcR2A-1 MFI: if significant compared to control is unclear from the graph.

FIGURE S2 how was correlation determined? Are these Rs and what is the P value? The legend is non described. What is the statistic? Is this corrected for multiple comparison?

FIGURE S3 arrow for neutrophil and macrophages infiltration may help.

Reviewer #2:

Remarks to the Author:

Thank you for asking me to review this manuscript on an RBD trimer vaccine

What are the major claims of the paper?

The manuscript was well written and shows that the trimer vaccine at doses lower than monomer can induce a strong neutralising antibody response and protect against infection and pathology in two animal models when the alum-3M-052 adjuvant is added. Previous published work has shown that monomer, dimer and ~60-mer RBD vaccines can be immunogenic and protective (referenced by the authors), and established the principal that multimerization improves the antibody response at low antigen dose. This paper shows this is true also of a trimer, and is enhanced by the alum-3M-052 mixed adjuvant. The properties of this adjuvant have also been described before (referenced by the authors).

Is the work convincing, and if not, what further evidence would be required to strengthen the conclusions?

The work is clearly well done and reasonably convincing. One key element I could not find was the yield of trimer/L of tissue culture, and some discussion of whether the production of this trimer in bulk was likely to reduce the costs of production of a practical vaccine. Figure 1 C only shows a western blot of the minor trimer peak 3, which suggests the yields of trimer might be very low compared to the standard expression of monomer (100-200mg/L) using similar methods in standard laboratory tissue culture. Also looking at Fig 2 the difference in neutralising titres between monomer and trimer are not massive. Also, in Figure 3 the differences in titres between animals vaccinated with Alum versus the Alum-3M-052 do not stand out. One wonders if given the likely greater yield of monomer RBD over trimer, if just employing a bigger dose of a monomer with a standard adjuvant would be as efficacious as the trimer with Alum-3M-052, and cheaper to produce.

On a more subjective note, do you feel that the paper will influence thinking in the field?

No. There was no new principle or insight introduced in this paper. It is a solid piece of research that extends to a minor degree what is now known.

We would also be grateful if you could comment on the appropriateness and validity of any statistical analysis, as well the ability of a researcher to reproduce the work, given the level of detail provided.

I am not a good enough statistician to criticise. Yes, the detail is sufficient to reproduce the work.

Response to Reviewers' comments:

We thank the Reviewer's for carefully reading the manuscript and providing insightful comments. We have revised the manuscript to address all comments and concerns and hope the revised manuscript will be acceptable for publication. Below is our point-by-point response in RED text.

Response to Reviewer #1:

In this manuscript Routhu and colleagues describe the immunogenicity and efficacy of RBD vaccine as a monomer or a trimer in mice and test the trimer adjuvanted with alum or TLR7/8 agonist formulation alum-3M-052 in monkeys. The trimeric adjuvanted vaccines protect animals from VL and disease and mice receiving the alum -3M-052 have higher ab titers including neutralization. In monkeys, the 2 adjuvants are comparable in protection while differing in some instances after infection, for example in the frequency of innate cells. The authors comment that the possible lack of difference may be due to the dose or the sample size.

Overall, the draft needs careful revision for typos, repetitions, and for improving descriptions of the data in a scientific appropriate manner (examples: "quite strong" for the antibodies titers "strong protection"; In particular, protection needs to be defined.

We sincerely thank the reviewer for carefully reading the manuscript and providing constructive inputs. We apologize for these errors and have corrected the sentences to be more precise throughout the text.

The reason why the alum 3M-052 was proposed is to study its potential in enhancing durability of responses, yet duration of such responses was not studied in this study. It would be helpful to include analysis of B cells in bone marrow (LLPC), if available, or plasmablasts in lymph nodes to look for differences suggesting longevity of responses in alum vs. alum-M-052. We agree with the reviewer that it would have been helpful to have the data for long-lived plasma cells (LLPCs) in bone marrow or plasmablasts in the LN. However, a recent report using nanoparticle formulation of 3M-052 in rhesus macaques showed that LLPCs appear at very low frequency (10 spots/million cells) even at 5 weeks after the 2nd immunization and a 3rd immunization is required to see good frequencies (1). In our study we challenged animals 4 weeks after the 2nd immunization. Given this prior data and to minimize the stress on animals before challenge we did not collect bone marrow and LN after vaccination and hence, we are unable to provide these data. However, it is conceivable that the superior innate activation observed in the alum-3M-052 group could lead to induction of long-lived antibody responses in this group. We hope the reviewer can understand these limitations.

In discussion:

There is a lack of contextualization in previously published studies on correlates in protection in humans and macaques (innate and adaptive) and particularly on similar vaccines tested in the same models. In particular, discussion should include

1) A more inclusive comparison with similar vaccines (RBD immunogens) tested in same models with different results in macaques and mice (example: Tan, HX. Ref15);

We have expanded discussion to include recent studies that used RBD protein as an immunogen.

2) Discussion about innate responses as correlates of risk in macaques and humans (Fahlberg MD Nat Commun. 2020 Nov 27;11(1):607, Singh, D.K, Nat Microbiol 6, 73–86) (Wilk, A. J. et al. A single-cell atlas of the peripheral immune response in patients with severe COVID-19. Nat. Med. 26, 1070–1076 (2020). Would help explaining the analysis and findings in Figure 4.

We have revised the discussion to include findings from the studies mentioned by the Reviewer. With respect to explaining the analysis and findings in Fig. 4, if the discussion is about activation of innate cells following vaccination and how they relate to infection risk following challenge we don't think this would be a problem since the innate activation following vaccination is transient (1).

Moreover

-“They also showed that RBD trimer vaccination protects from lung pathology induced by vaccination”. not clear:

We corrected ‘vaccination’ to SARS-CoV-2 infection at the end of sentence.

-“Vaccinations induced Th1-biased CD4 T cell response which is important for protection against SARS-CoV-2”. Reference needed

Thanks for pointing this out. We have added references.

-“The protection was mainly evident in the lower respiratory tract (lungs) and to a lesser extent in the upper respiratory tract (nasopharynx)”. Define protection and figures when this is shown.

-“The viral load is more variable in the nasopharynx and larger group sizes are required to make definitive conclusions about vaccine efficacy in this compartment” contradicts what just stated, not clear.

We modified these sentences to clarify protection from virus replication and added figure number.

Results

Fig 2D Is the correlation significant when the 2 groups are analyzed separately? It appears that the difference between the 2 groups included is what is driving it.

The correlations are still significant when the two groups are analyzed separately but the p values are less strong as one would expect. For the Alum group $p=0.02$, $r=0.8$; for Alum-3M-052 group $p=0.05$, $r=0.7$.

Methods:

SARS COV2 challenge of macaques, which virus?

We used 2019-nCoV/USA_WA1/2020 (A.1 lineage) strain. This was obtained from BEI resources amplified in VeroE6 cells (ATCC) and sequenced. We have added the details to the methods section.

Figures

-Figure 2 should include the timing under figures (prime and boost). What is the statistical analysis for B, C, E, F, G, H I is it corrected for multiple comparison? Why there is only 3 animals for the viral load Fig 2I? There are N= 5 animals described yet there are 6 animals per group in Figure 2D. Figure 2F: what is the significance here? With the control?

Timing – We have updated this figure to include analysis time point for all measurements.

Re the statistical test – we used a two-tailed nonparametric Mann–Whitney rank-sum test to compare the differences between the groups and timepoints, and a p -value of less than 0.05 was considered significant. The p values were not corrected for multiple comparisons. However, after Bonferoni correction the threshold cut-off of p value is 0.025 and all p values are still below this threshold wherever applicable. We have clarified this in the figure legend.

Why 3 animals in Fig. 2I – As described in the text, of the 5 vaccinated mice in each group we used 3 mice to measure viral titer in the lungs at Day 2 and two mice to define the weight loss at Day 5. Because of this, we combined animals vaccinated with trimer and monomer for each adjuvant, which gave us a total of 6 animals/adjuvant and compared all alum-3M-052 vaccinated animals with alum vaccinated animals.

Six animals/group shown in Fig. 2D – looks like there is some confusion. We showed only 5 animals/group.

Significance shown in Figure 2F – It is true that the p value shown is compared to controls. We have clarified this in the figure.

-Figure 3: Are the stars representative of significance with controls? Are these corrected? What is the color of the legend corresponding to in F? and it is that Person test correlaplot?

The stars are representative of significance with their respective prevaccination bleeds. The p values were calculated using a two-tailed nonparametric Mann-Whitney test. The p -value of less than 0.05 was considered significant. These were not corrected for multiple comparisons.

Figure 3F: The color refers to the r value scale shown on the right. In addition, we included the actual r value for each comparison so that it is easy for the reader to know actual value. The correlations were defined using the Spearman rank test. The p values were represented with stars i.e., $*p < 0.05$, $**p < 0.01$, and $***p < 0.001$.

We have updated the Figure 3 legend to incorporate these details.

FIGURE S1: 3 groups presented. Statistical differences with what? FcR2A-1 MFI: if significant compared to control is unclear from the graph.

We apologize for the mistake and we have corrected it now. The p values refer to statistical difference between pre vaccination and 2 weeks post booster vaccination time points for each group. We have now revised the figure legend to include these details.

FcR2A-1 MFI – It is true that the MFI values for Fc γ R2A-1 in the alum group are barely higher than the pre vaccination values. However, the difference is statistically significant. We agree with the Reviewer that the difference is negligible and because of this we removed the p value from the graphs.

FIGURE S2 how was correlation determined? Are these Rs and what is the P value? The legend is non described. What is the statistic? Is this corrected for multiple comparison? We apologize for the lack of details. The correlation matrix was generated between Day 2 viral loads (BAL and Nasal) and respective antibody specificity or function. The color refers to r value scale shown on the right. The number in each cell indicate the actual r value. The Spearman rank test was used to perform correlation analysis. None of the correlations were statistically significant. Because of this we clearly stated in the text that we observed only moderate associations ($r > 0.5$, $p < 0.1$) for some functions. The lack of significant correlations could be due to small sample size. We updated the figure legend with this information.

FIGURE S3 arrow for neutrophil and macrophages infiltration may help.

We added Figure S4 with representative 20x images for two control animals and added arrows to show neutrophils and macrophages.

Response to Reviewer #2:

The manuscript was well written and shows that the trimer vaccine at doses lower than monomer can induce a strong neutralising antibody response and protect against infection and pathology in two animal models when the alum-3M-052 adjuvant is added. Previous published work has shown that monomer, dimer and ~60-mer RBD vaccines can be immunogenic and protective (referenced by the authors) and established the principal that multimerization improves the antibody response at low antigen dose. This paper shows this is true also of a trimer and is enhanced by the alum-3M-052 mixed adjuvant. The properties of this adjuvant have also been described before (referenced by the authors).

We thank the Reviewer for carefully reading the manuscript and positive comments.

The work is clearly well done and reasonably convincing. One key element I could not find was the yield of trimer/L of tissue culture, and some discussion of whether the production of this trimer in bulk was likely to reduce the costs of production of a practical vaccine. Figure 1 C only shows a western blot of the minor trimer peak 3, which suggests the yields of trimer might be very low compared to the standard expression of monomer (100-200mg/L) using similar methods in standard laboratory tissue culture.

We thank the reviewer for the comment. In this transient transfection system the yield we get from the peak 3 which is specific for RBD trimer is about 2.2 mg/liter culture. Using the same

conditions in our hands the yield for RBD monomer with His tag is about 7.6 mg/liter. The higher yield we observed for RBD monomer could be due to the use of high affinity his-tag based purification system. However, we used binding to ConA followed by size exclusion chromatography to purify the RBD trimer protein, which would decrease yield. We have plans to improve the yield of trimer by tagging it to Strep-Tag II plus His tag and HRV 3C cleavage system as reported by Daniel Wrapp et al., Science, 2020. In addition, making a stable cell line expressing the trimer should also markedly improve the yield. We have updated the text with the protein yield and potential to improve it.

Also looking at Fig 2 the difference in neutralising titres between monomer and trimer are not massive. Also, in Figure 3 the differences in titres between animals vaccinated with Alum versus the Alum-3M-052 do not stand out. One wonders if given the likely greater yield of monomer RBD over trimer, if just employing a bigger dose of a monomer with a standard adjuvant would be as efficacious as the trimer with Alum-3M-052, and cheaper to produce.

It is important to note that although both monomer and trimer induced comparable binding antibody titer (Fig. 2G) the neutralizing antibody titer is lower in the monomer compared to trimer (Fig. 2H). This raises the possibility that the functional quality of the antibody response induced by trimer likely superior to monomer. With respect to the yield differences between the monomer and trimer proteins, as explained above the purification methods need to be matched to make any conclusions.

There was no new principle or insight introduced in this paper. It is a solid piece of research that extends to a minor degree what is now known.

This would be the first report showing the immunogenicity and efficacy of RBD trimer protein with alum-3M-052 as an adjuvant in mice and NHPs. Given the potential of this adjuvant to induce long lasting antibody responses in NHPs, the results from this study will help to advance protein-based vaccines for COVID-19.

References:

1. Kasturi SP, Kozlowski PA, Nakaya HI, Burger MC, Russo P, Pham M, Kovalenkov Y, Silveira EL, Havenar-Daughton C, Burton SL, Kilgore KM, Johnson MJ, Nabi R, Legere T, Sher ZJ, Chen X, Amara RR, Hunter E, Bosinger SE, Spearman P, Crotty S, Villinger F, Derdeyn CA, Wrammert J, Pulendran B. 2017. Adjuvanting a Simian Immunodeficiency Virus Vaccine with Toll-Like Receptor Ligands Encapsulated in Nanoparticles Induces Persistent Antibody Responses and Enhanced Protection in TRIM5alpha Restrictive Macaques. J Virol 91.

Reviewers' Comments:

Reviewer #1:

Remarks to the Author:

The authors have addresses the issues raised and substantially improved the manuscript.

Reviewer #2:

None